# A stochastic mechanism drives fast substrate translocation in the AAA+ machine ClpB

Remi Casier [1], Dorit Levy[1], Inbal Riven[1], Yoav Barak[2] & Gilad Haran [1] ✉

How biological machines harness ATP to drive mechanical work remains a crucial question. Structural studies of protein-translocating AAA+ machines proposed a coupled and sequential translocation process, whereby ATP hydrolysis events lead to short threading steps. Yet, direct real-time observation of these events remains elusive. Here, we employ single-molecule FRET spectroscopy to track substrate translocation through ClpB, a quality control AAA+ machine. We isolate ClpB and its substrate within lipid vesicles and find that translocation events, while dependent on ATP, take milliseconds, much faster than ATP hydrolysis times. Surprisingly, the translocation rate depends weakly on temperature and ATP concentration. Using three-color FRET experiments, we find that translocation events can occur bidirectionally but are not always complete. Replacing ATP with the slowly hydrolysable analog ATPγS abolishes both rapid translocation and directionality. These results indicate a fast, stochastic Brownian-motor-like mechanism, redefining how ATP is coupled with mechanical action in AAA+ machines.

Proteins of the ubiquitous AAA+ (ATPases associated with diverse cellular activity) superfamily participate in essential cellular activities, including protein disaggregation and degradation, DNA replication and recombination, organelle biogenesis, and cellular remodeling[1–3]. Despite their remarkable functional diversity, they typically assemble into ring-like hexamers and share conserved nucleotide-binding domains (NBDs) that couple ATP turnover to mechanical work[3]. However, how ATP energy is utilized to promote the activity of these proteins remains an active area of study. In recent years, cryogenic electron microscopy has yielded numerous 3D structures of AAA+ machines, with a particular focus on those that unfold or disaggregate client proteins and then translocate them through their central lumen for protein quality control activities. Rather than adopting a fully symmetric ring, many of the AAA+ assemblies were found to form a spiral-like configuration[4–7]. These observations, based on static structural information, have led to the proposed hand-over-hand model for substrate translocation. In this model, the top subunit sequentially moves downwards following ATP hydrolysis, pulling the substrate protein in a *power-stroke-like* manner[4,8–13]; each ATP hydrolysis event directly generates a single forceful translocation step of two amino-acid (*aa*) residues[8]. However, complementary biochemical and

biophysical studies point towards a more stochastic and rapid mechanism. For instance, experiments on the AAA+ machines ClpA and ClpX demonstrated that they can function in translocating proteins even when up to three of their subunits are defective in ATP hydrolysis[14,15], suggesting that full coordination is not strictly required. Further, optical tweezer experiments on the AAA+ disaggregation machine ClpB[16,17] found that it could translocate more than 500 *aa*'s in a second[18], outpacing the ATP hydrolysis rate by more than two orders of magnitude. Additional studies have also pointed to the viable possibility of partial substrate threading by ClpB[19]. These observations underscore the need for direct, real-time monitoring of substrate translocation to obtain decisive evidence for the mechanism of action of these machines.

Here, we address this challenge by using single-molecule Förster resonance energy transfer (smFRET) experiments to directly track, in real-time, the motion of a substrate protein through the lumen of ClpB. As a model substrate, we use κ-casein, which has a high degree of intrinsic disorder[20,21]. This property allows us to bypass the unfolding requirement of structured proteins, which has been studied elsewhere (see e.g. refs. 22–24) and thereby isolate the translocation process itself, which is expected to be faster and is a central component of

[1]Department of Chemical and Biological Physics, Weizmann Institute of Science, Rehovot, Israel. [2]Chemical Research Support, Weizmann Institute of, Rehovot, Israel. ✉e-mail: gilad.haran@weizmann.ac.il

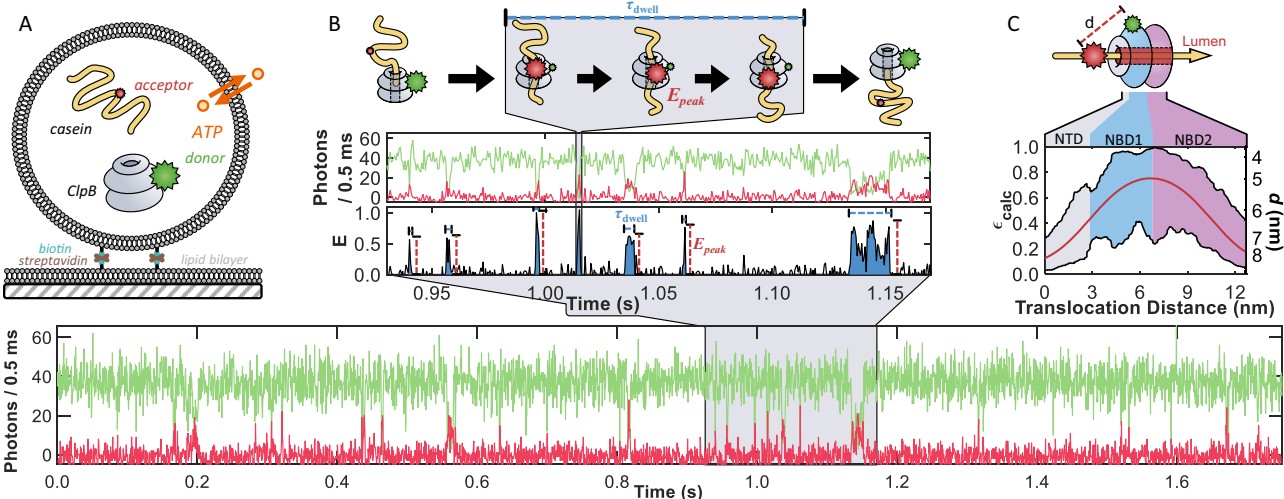

**Fig. 1 | Observation of multiple fast events of translocation through the lumen of ClpB facilitated by porous vesicles. A** Schematic overview of a ClpB and substrate-loaded liposome (diameter ca. 120 nm) immobilized on a glass-supported lipid bilayer through a biotin-streptavidin linkage. Only liposomes containing individual copies of a fluorescently labeled ClpB hexamer (grey, with a green donor, Cy3b, on residue 359) and casein (yellow, with a red acceptor, CF660R) were included in the analysis. Liposomes remained permeable to ATP (orange) at 22.5 °C, ensuring replenishment of the ATP supply consumed by ClpB. **B** A sample fluorescence trajectory (bottom) of a ClpB-casein pair inside a liposome in the presence of 2 mM ATP (donor signal, green, and acceptor signal, red). A zoomed-in region of the trajectory (middle) depicts several short events of ClpB

and casein interaction. These events are seen as the sudden appearance and disappearance of spikes in the apparent FRET efficiency, $E_{app}$ ($= N_A/(N_D + N_A)$), where $N_D$ and $N_A$ are donor and acceptor counts, respectively. Each spike reaches a maximum value of $E_{peak}$ and is characterized by a time $\tau_{dwell}$. A cartoon (top) illustrates a translocation event of casein through the lumen of ClpB. **C** A map of the expected FRET efficiency ($\varepsilon_{calc}$) for an acceptor-labeled segment as it passes through ClpB with a donor on the NBD1. The solid red line represents the centerline passing directly through the pore, while the black lines represent the boundaries of the pore. The colored regions correspond to the N-terminal domain (NTD, grey) and the two nucleotide-binding domains (NBD1 and NBD2, blue and purple, respectively). The secondary y-axis denotes the inter-dye distance (d).

ClpB-mediated disaggregation and AAA+ machine interaction with proteins in general. Studying this step in isolation facilitates probing the fundamental mechanism underlying substrate movement through the central pore. Our experiments are performed on ClpB molecules isolated with casein substrates within surface-immobilized lipid vesicles. We find that ATP is required for translocation, yet it only weakly modulates its velocity. Instead, it primarily regulates the frequency and directionality of motion. Three-color smFRET measurements show that substrates move bidirectionally through the lumen with a strong forward preference, and that replacing ATP with ATPγS abolishes both rapid movement and directionality. These results lead us to propose a stochastic, ATP-dependent translocation mechanism in which the consumption of ATP regulates the directionality of substrate diffusion within the lumen. This *Brownian-motor-like* mechanism[12] enables efficient translocation, achieving complete threading of an unfolded substrate with the consumption of only one or two ATP molecules.

## Results
### Observing substrate translocation events by ClpB
To gain insight into the molecular mechanism of ClpB (*Thermus thermophilus*), we utilized real-time smFRET to monitor the translocation of the model substrate κ-casein, an unstructured protein[20,21]. We inserted individual copies of the fluorescently tagged proteins inside surface-tethered lipid vesicles (Fig. 1A), which afforded several distinct advantages[25]. The confinement into ca. 120 nm diameter vesicles (Supplementary Fig. 1) created a high local protein concentration (ca. 2–3 μM), which facilitated the bimolecular interactions between the weakly associating species ($K_{D,app}$ (caein) = 0.6 ± 0.1 μM, see below and Supplementary Fig. 2), while still allowing the proteins to diffuse freely (Supplementary Fig. 3). The immobilized vesicles also provided extended observation times for individual ClpB molecules, allowing us to monitor repeated translocation events. The liposomes were prepared from 1,2-dimyristoyl-sn-glycero-3-phosphocholine (DMPC), a

lipid that is near its gel-to-liquid crystalline transition at room temperature (22.5 °C), allowing for the permeation of ions such as ATP[26,27]. The rapid and reversible permeability of the membrane (see *liposome permeability* and Supplementary Fig. 4 in the Supplementary Information) facilitated the maintenance of constant ATP levels.

We labeled casein on a native cysteine and ClpB on a cysteine inserted into the nucleotide-binding domain 1 (NBD1) of only one of its six subunits (Supplementary Figs. 5 and 6). A sample fluorescence trajectory containing multiple events of interaction between ClpB and casein in the presence of 2 mM ATP is shown in Fig. 1B. Each liposome contained a single labeled casein molecule, and the fluorescence trajectory therefore reports multiple, repeated interaction events between the same casein and ClpB. Most of the time, the trajectory shows no energy transfer between the dyes, as ClpB and casein diffuse independently within the liposome. However, there are sudden, brief appearances of high FRET efficiency, indicating that casein has entered the lumen of ClpB (blue, Fig. 1B inset). The peaks in FRET efficiency were characterized by the temporal length of the events ($\tau_{dwell}$) and the maximum FRET efficiency values reached ($E_{peak}$). As illustrated at the top of Fig. 1B, the 76 nm (= 190 $aa$ × 0.4 nm per $aa$[28]) contour length of casein far exceeds the ca. 13 nm length of ClpB's lumen (hexameric model based on PDB 1QVR[29]) and the 6 nm Förster radius ($R_0$) of the dye pair. Therefore, $\tau_{dwell}$ does not correspond to the translocation process of the entire chain of casein but rather to a temporal segment during which the acceptor-labeled portion of casein passes through ClpB. To help illustrate this, Fig. 1C presents a calculated map of the expected FRET efficiency ($\varepsilon_{calc}$) as the acceptor traverses the central pore of ClpB. As the fluorescently labeled residue enters the N-terminal domain (NTD), $\varepsilon_{calc}$ is less than 0.3, depending on the exact location within the cross-section of the 2 – 3 nm diameter pore. As the dye travels deeper into the lumen, $\varepsilon_{calc}$ increases, potentially reaching a value close to unity around the interface between the NBDs. $\varepsilon_{calc}$ then decreases as the labeled segment exits the trans side past the NBD2.

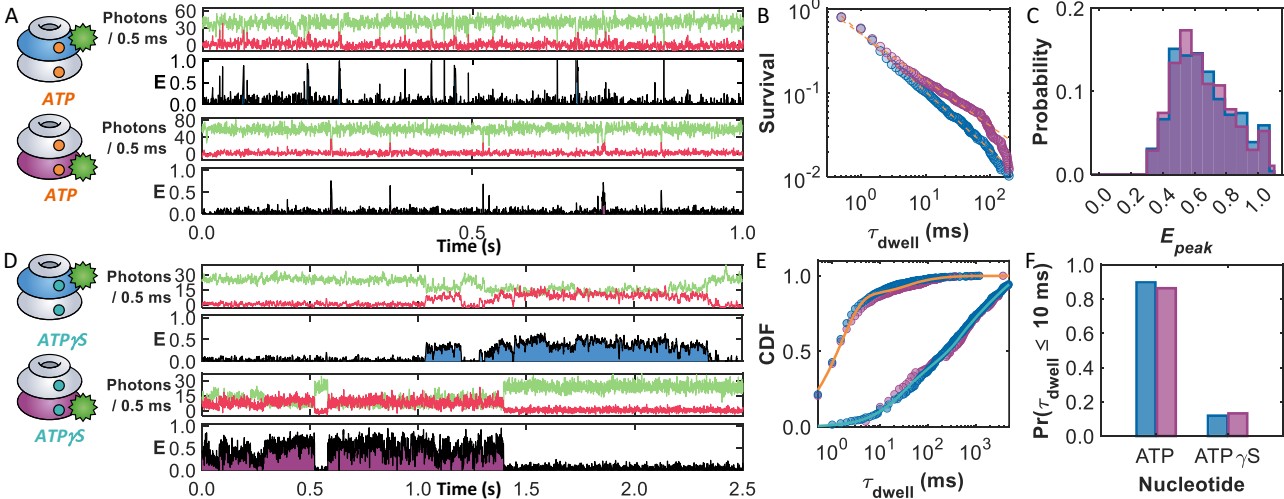

**Fig. 2 | ATP, but not ATPγS, sustains rapid substrate processing.** Sample trajectories in the presence of (**A**) 2 mM ATP or (**D**) 2 mM ATPγS. ClpB was labeled with a donor either on residue 359 in NBD1 (top trajectory) or residue 771 in NBD2 (bottom trajectory). **B** The survival probability functions of the events for NBD1- (blue) and NBD2- (purple) labeled ClpB are very similar, with power-law fits (dashed lines) yielding powers of 1.69 ± 0.02 and 1.55 ± 0.02, respectively. **C** Histogram of the peak FRET efficiency, $E_{peak}$, for the events in (**B**). **E** Cumulative distribution function (CDF) of the event times for the two labeling locations in the presence of ATP and ATPγS. The solid lines represent triexponential fits of the dwell times in the presence of (orange) ATP and (teal) ATPγS, providing average values of 21.8 ± 0.3 and 749 ± 4 ms, respectively. **F** The presence of ATPγS suppresses the fraction of short (<10 ms) events in both labeling locations, demonstrating that the fast translocation events are ATP-dependent.

Thus, $\tau_{dwell}$ values represent the time that it takes the labeled residue to traverse a segment of the ClpB pore corresponding to a length equivalent to ~18 $aa$ (0.4 nm per[28]) of the substrate (where $\varepsilon_{calc} > 0.5$), and $E_{peak}$ corresponds to the minimum proximity between the two labels during each translocation event, rather than the motion of the entire polypeptide chain.

### smFRET reveals ultrafast translocation of casein

From the sample trajectories given in Figs. 1B and 2A (ClpB labeled on the NBD1), it can be readily seen that the events occur very quickly, lasting only a few milliseconds. The 1535 events measured across 166 trajectories resulted in a power-law distribution of dwell times, $P(\tau_{dwell}) \sim \tau_{dwell}^{-a}$, with a power $a = 1.69 \pm 0.02$ (Fig. 2B). Although the $\tau_{dwell}$ values spanned several orders of magnitude, the majority (89%) of the events occurred in less than 10 ms. Power laws with a power less than two do not have well-defined means[30], so an estimate of the average dwell time ($\langle\tau_{dwell}\rangle$) in our experiments was obtained from an exponential fit to the short (<10 ms) translocation times. The fit revealed a $\langle\tau_{dwell}\rangle$ value of only 1.6 ± 0.1 ms, nearly 3 orders of magnitude faster than the ATP hydrolysis time of the machine (1.14 ± 0.03 ATP/s, Supplementary Fig. 5). To preclude the possibility that the short $\tau_{dwell}$ values were due to acceptor blinking (a photophysical artifact), we verified that the acceptor remained fluorescently active throughout the entire trajectory (Supplementary Fig. 7). We also verified that, as demonstrated before, ClpB retained its hexameric assembly (Supplementary Fig. 5) under these conditions, necessitating that casein can only enter ClpB through the lumen openings. Given that the millisecond $\tau_{dwell}$ values are much longer than the timescale of the dynamics of an unfolded protein chain (10's–100's ns[31,32]), there is ample time for the chain segment to explore the entire cross-sectional area of the pore during translocation. Even if interactions with ClpB slow these motions, they would remain orders of magnitude faster than our experimental time resolution of milliseconds. In addition, the flexibility of the dyes, linkers, and disordered chain further contributes to temporal and spatial averaging of the observed FRET efficiencies. We acknowledge that our $\varepsilon_{calc}$ calculation, based on slicing the ClpB structure into planes and measuring distances from a fixed labeling site, does not explicitly account for the flexibility of casein, the dyes, or

their linkers. Instead, $\varepsilon_{calc}$ serves as a simplified geometric reference, providing an approximate measure of the average substrate position within the lumen rather than a fully quantitative structural model. For clarity, we present $\varepsilon_{calc}$ values corresponding to a chain passing through the center of the pore (red center line in Fig. 1C), which offers a consistent framework for comparing experimental and calculated FRET efficiencies without requiring detailed dynamic simulations.

Comparing the $E_{peak}$ values (Fig. 2C) to the $\varepsilon_{calc}$ map (Fig. 1C), it becomes clear that the moderate $E_{peak}$ values of ca. 0.5 correspond to instances in which the tagged residue of casein only begins to enter the pore (> 3 nm depth). If entry is via the NTD, it indicates that casein has reached the NBD1 but has not made it past the first of three sets of flexible loops extending into the lumen (pore loops 1, PL1, located at a depth of ca. 3.4 nm). On the other hand, those 45% of the events with $E_{peak}$ values ≥ 0.7 must originate from cases where casein reaches deep into the lumen of ClpB, near the interface between the two NBDs. This could be accomplished by casein entering through the NTD ring, reaching NBD1, and passing through (at least) PL1 and PL2 (the depth of the latter is ca. 4.4 nm), corresponding to a translocation depth of at least ca. 5 nm into the lumen.

To examine whether translocation events culminated in a complete passage through the lumen of ClpB, we repositioned the donor molecule to the NBD2 of ClpB (Fig. 2A, bottom). In this position, high FRET efficiency events can only occur if casein passes through the NBD2 ring (Supplementary Fig. 8). The sample trajectory shown in Fig. 2A bottom (in purple) is remarkably similar to those obtained with the NBD1-labeled ClpB (Fig. 2A top). In fact, the 1214 events measured across 137 trajectories displayed a similar power-law distribution ($a = 1.55 \pm 0.02$, Fig. 2B), and identical average event time $\langle\tau_{dwell}\rangle = 1.6 \pm 0.1$ ms) and $E_{peak}$ values (Fig. 2C) compared to those of the NBD1-labeled protein. Additional sample trajectories for both labeling locations are shown in Supplementary Figs. 9 and 10, and a summary of statistics is provided in Supplementary Tables 1 and 2. The high $E_{peak}$ events in the histogram (Fig. 2C), constituting 45% of all events, necessitate that the acceptor-labeled segment has completed passage through the NBD2 and through the third ring of pore loops (PL3, depth of ca. 7.8 nm). The low $E_{peak}$ values of ca. 0.5 may correspond to events where casein has made its way past the first two PL rings but has not

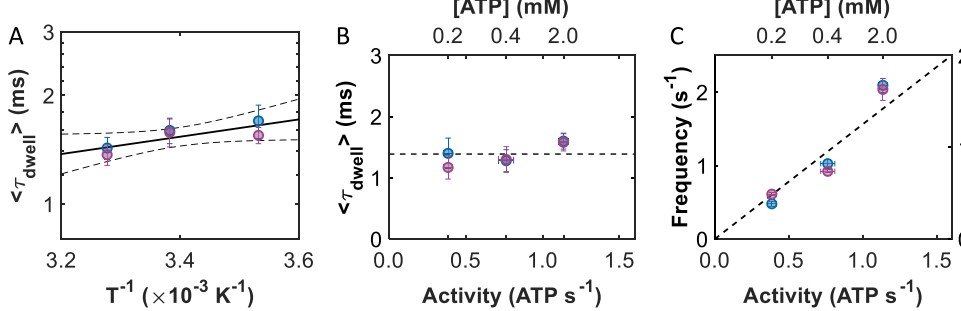

**Fig. 3 | Temperature and ATP concentration dependence measurements support a ratcheting mechanism. A** Arrhenius plot (logarithm of $\langle\tau_{dwell}\rangle$ vs. the reciprocal of temperature) of the average dwell times for NBD1- (blue) and NBD2- (purple) labeled ClpB. Linear regression (solid black line) returns an activation energy of $1.87 \pm 0.69$ $k_B$T. The 95% confidence intervals are provided as dashed lines. **B** $\langle\tau_{dwell}\rangle$ values for NBD1- (blue) and NBD2- (purple) labeled ClpB as a function of ATPase activity. The dashed line represents the average dwell time of $1.4 \pm 0.2$ ms. The ATPase activity of ClpB (bottom axis, Supplementary Fig. 5) was controlled by varying the ATP concentration (top axis) in the smFRET experiments. **C** The frequency of events for NBD1- (blue, left axis) and NBD2- (purple, right axis) labeled ClpB as a function of ATPase activity. Linear regressions return $1.65 \pm 0.18$ and $1.29 \pm 0.15$ events per hydrolysed ATP based on the bulk ATPase activity. Note the non-linear scale in ATP concentration in panels B and C, top axes. Data are presented as mean values +/− the standard deviation (SD); values are summarized in Supplementary Tables 1 and 2.

completed translocation across NBD2. The similar $\langle\tau_{dwell}\rangle$ values for NBD1- and NBD2-labeled ClpB suggest that the longitudinal motions (estimated to correspond to distances of 8.1 and 6.2 nm, respectively, based on the region in the $\varepsilon_{calc}$ maps where $\varepsilon_{calc} > 0.5$) occur rapidly across both NBDs. While these fast events only represent snapshots of motion corresponding to contour lengths of ca. 16–20 *aa*'s along casein, the overwhelming frequency of these short events points towards a rapid translocation mechanism spanning both NBDs.

### Millisecond translocation is ATP-dependent
The threading velocity through ClpB (~ 20 *aa*'s in 2 ms) estimated from the above experiments is much faster than expected based on its overall ATPase activity ($1.14 \pm 0.03$ ATP/s, Supplementary Fig. 5), begging the question as to whether the events measured here truly represent ATP-dependent translocation. While it would be tempting to simply remove ATP to abolish ClpB activity, the hexameric assembly of ClpB necessitates the presence of ATP or an analog[33]. The slowly hydrolysable ATPγS (adenosine-5'-o-3-thio-triphosphate) is known to hamper the translocation activity of ClpB, resulting in a *holdase*-like behavior[34]. We, therefore, tested events of interaction between ClpB and casein in the presence of 2 mM ATPγS, collecting 337 events from 147 molecules with the NBD1 labeling location and 227 events from 113 molecules with the NBD2 labeling location. Notably, compared to the same labeling locations in the presence of ATP, there was a > 4-fold reduction in the average number of events (compare to >1200 events for a similar number of molecules with ATP, Supplementary Table 2). Figure 2D presents sample trajectories for both labeling locations on ClpB (NBD1 in blue, NBD2 in purple) in the presence of 2 mM ATPγS, illustrating a substantial increase in substrate dwell time (see also Supplementary Figs. 11 and 12). A cumulative dwell time histogram of both labeling locations in the presence of ATP and ATPγS is given in Fig. 2E. While the presence of ATP leads to nearly 90% of the events occurring in less than 10 ms (Fig. 2F), ATPγS suppresses these short events, reducing their frequency to <15%. Fitting the cumulative distribution of dwell times (solid lines in Fig. 2E, Supplementary Table 3) revealed that the weighted average overall dwell time increased from $21.8 \pm 0.3$ to $749 \pm 4$ ms when the nucleotide was switched from ATP to ATPγS, corresponding to an increase by a factor of ~34.

In support of these findings, the apparent dissociation constants ($K_{D,app}$) of casein in the presence of both nucleotides were determined by fluorescence anisotropy (Supplementary Fig. 2). A simple bimolecular binding curve revealed that ATPγS decreased $K_{D,app}$ (= $20 \pm 10$ nM) by a factor of 30 compared to that of ATP (= $600 \pm 10$ nM). Importantly, due to the active translocation and

disengagement by ClpB, these $K_{D,app}$ values do not represent the true dissociation constant but rather depend on the interaction times with casein. Therefore, the agreement between the 30-fold increase in dwell time and apparent equilibrium constants supports the conclusion that the events probed in the single-molecule experiments indeed report on the translocation activity of ClpB. Overall, the presence of ATP generates fast translocation events that last only a few milliseconds. Despite these events occurring on a timescale much faster than the bulk ATPase activity, the presence of ATPγS hinders them considerably, suggesting that with this nucleotide, substrates are not quickly released by the machine but rather dwell inside for long periods of time.

### Translocation times depend weakly on temperature
To glean further insight into the mechanism of translocation, the temperature dependence of the measured events was investigated by repeating the smFRET experiments at two additional temperatures, 10 and 32 °C (see *Temperature-dependent smFRET experiments* for more details). At each temperature, over 60 ClpB molecules were sampled, resulting in the observation of hundreds (>500) of events (Supplementary Table 1). Sample trajectories are provided in Supplementary Fig. 13. For both labeling locations, the impact of temperature on $\tau_{dwell}$ was surprisingly small, leading to similar power laws, $E_{peak}$ histograms, and fraction of short events (Supplementary Fig. 14, Supplementary Table 2). In fact, only a slight increase in the $\langle\tau_{dwell}\rangle$ value from 1.4 to 1.7 ms was observed as the temperature decreased from 32 to 10 °C (Supplementary Table 2). An Arrhenius plot combining data from both labeling locations (Fig. 3A) revealed a very low activation energy for translocation of $E_A = 1.87 \pm 0.69$ $k_B$T. For comparison, the molecular motors kinesin and actomyosin, which consume ATP to '*walk*' along microtubules, exhibit significantly larger energy barriers of 20 and 40 $k_B$T, respectively[35,36]. These classically established *power-stroke* machines exhibit large barriers due to large conformational changes driving the mechanical work[12,13,37], as would also be expected for ClpB if ATP were directly coupled with discrete translocation steps. Instead, the substantially lower activation energy measured for ClpB demonstrates that it must utilize a mechanism for translocation that does not rely on large structural shifts, such as a *Brownian ratchet*[12,38,39] (see "Discussion").

### Event frequency directly correlates with bulk ATPase activity
At a saturating concentration of ATP (2.0 mM, Supplementary Fig. 5), the bulk casein-stimulated ATPase activity of ClpB is only $1.14 \pm 0.03$ ATP/s. How then is the slow consumption of ATP related to the two to

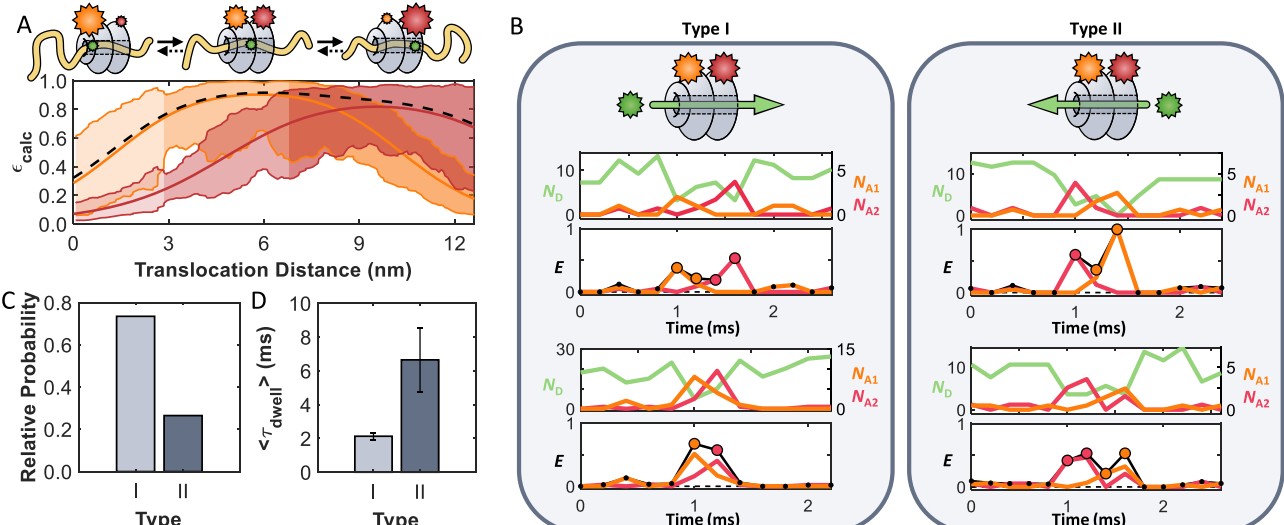

**Fig. 4 | Translocation of casein in both the forward and backward directions is observed in three-color experiments. A** Map of the calculated FRET efficiency as a function of location within the lumen of ClpB. Acceptors are located on the NBDs of ClpB (acceptor 1: AF647 ($R_O$ ~ 6.2 nm), orange, res. 176, and acceptor 2: CF680R ($R_O$ ~ 7.2 nm), red, res. 771). The thick orange, red, and black lines represent the calculated FRET efficiency through the center of the pore to acceptors 1 and 2, and the total FRET efficiency, respectively. The shaded areas represent the limits of the FRET efficiencies based on the boundaries of the pore. **B** Complete translocation events through ClpB. Type I – forward translocation events starting at the NTD and exiting past the NBD2. Type II – backward translocation events starting at the NBD2 and exiting past the NTD. Samples of the fluorescence of the donor (green, $N_D$) and acceptors (acceptor 1: orange, $N_{A1}$ and acceptor 2: red, $N_{A2}$) and apparent FRET efficiency ($E_{A1} = N_{A1}/(N_D + N_{A1} + N_{A2})$, $E_{A2} = N_{A2}/(N_D + N_{A1} + N_{A2})$, and overall apparent efficiency as the black trace $E_{app} = E_{A1} + E_{A2}$) are provided. Each point in time was assigned a color based on which acceptor was more prevalent (i.e., which acceptor experienced greater energy transfer from the donor, see *smFRET data analysis*) as indicated by the coloring of the data points in the FRET efficiency trace. **C** The fractions of complete translocation events in the forward (light grey, 438 events, 135 ClpB molecules) and backward (dark grey, 158 events, 135 ClpB molecules) directions when considering only complete translocation events, and **D** the average event times in the forward ($2.1 \pm 0.2$ ms) and backwards ($6.6 \pm 1.9$ ms) directions (Supplementary Fig. 20). Data are presented as mean values +/- SD, values are summarized in Supplementary Table 6.

three orders of magnitude faster translocation events? To begin exploring this question, we repeated single-molecule experiments under conditions where the ATPase activity of ClpB was reduced. By lowering the concentrations of ATP from 2.0 to 0.4 and 0.2 mM, the bulk ATPase activity was lowered from $1.14 \pm 0.03$ to $0.76 \pm 0.05$ and $0.39 \pm 0.03$ ATP/s, respectively (Supplementary Fig. 5). Translocation events from over 100 individual ClpB molecules were collected at each ATP concentration and labeling location (Supplementary Table 1), and sample trajectories are shown in Supplementary Fig. 15. Despite the significant reduction in ATPase activity with decreasing ATP concentration, we observed only small changes in the average dwell times (Fig. 3B), dwell time distributions, power-law exponents, and $E_{peak}$ histograms (Supplementary Tables 2 and 4, Supplementary Fig. 16). There was, however, a significant reduction in the frequency of events (= # of events/trajectory length) as the ATPase activity decreased. Exponential fits of the event frequency distribution (Supplementary Fig. 17) revealed that the frequency decreased linearly with decreasing activity (Supplementary Table 5). As seen in Fig. 3C, both labeling locations exhibited very strong correlations ($R = 0.98$ and $0.95$, respectively) between the event frequency and the bulk ATPase activity. The unanticipated result that decreasing ATP concentrations strongly affects event frequency, but only weakly affects event dwell times, reveals an important finding that ATP controls the recruitment of protein substrates by the machine. Earlier studies on hexameric AAA+ machines suggest that ATP may widen the entrance to the lumen[40,41], which may explain the facilitation of substrate recruitment. From the slopes of the results in Fig. 3C, we find that the average numbers of events per hydrolysed ATP molecule are $1.65 \pm 0.18$ and $1.29 \pm 0.15$ for NBD1- and NBD2-labeled ClpB, respectively. The larger-than-unity values are perhaps not unexpected, since it is possible that substrate recruitment by ClpB might be affected by the dwell times of

ATP products (ADP or $P_i$) on one or more of its 12 binding sites, rather than the ATP hydrolysis events themselves.

## Three-color smFRET studies reveal bidirectional translocation

While the two-color smFRET experiments demonstrated that casein can rapidly translocate past both NBDs, the question remained if these events represented complete translocation. Further, while previous studies suggested that the interaction with substrates starts at the N-terminus, there is no direct proof that translocation can take place only in this direction. To address these aspects, we designed a three-color FRET experiment whereby a donor dye (Cy3B) was attached to the substrate casein, and two acceptor dyes were positioned on ClpB. Using bioorthogonal labeling techniques to ensure location specificity, we labeled the NBD1 and NBD2 of ClpB with AF647 (acceptor 1) and CF680R (acceptor 2), respectively. The specific positions were chosen to maximize the sensitivity to the motion through the lumen while simultaneously minimizing unwanted energy transfer between the acceptors (Supplementary Fig. 18). A map of the $\varepsilon_{calc}$ in Fig. 4A depicts how the FRET efficiency between the dye on casein and each of the dyes on ClpB is expected to change as a function of position throughout the lumen. Given the relative locations of the dyes, acceptor 1 (orange) contributes more while casein traverses the NTD and NBD1, while energy transfer to acceptor 2 (red) becomes dominant as casein passes through the NBD2. We used pulsed interleaved excitation to collect 1445 signals of three-color events across 135 ClpB molecules with ATP, and 135 events across 110 molecules with ATPγS. Sample trajectories are provided in Supplementary Fig. 19. In the presence of ATP, the dwell-time distributions (decaying with a power law of $a = 1.68 \pm 0.2$), the fraction of short events (91% of the events persisted for less than 10 ms), and $E_{peak}$ (= max($E_{app}$)) values were all in close agreement with the two-color experiments (Supplementary

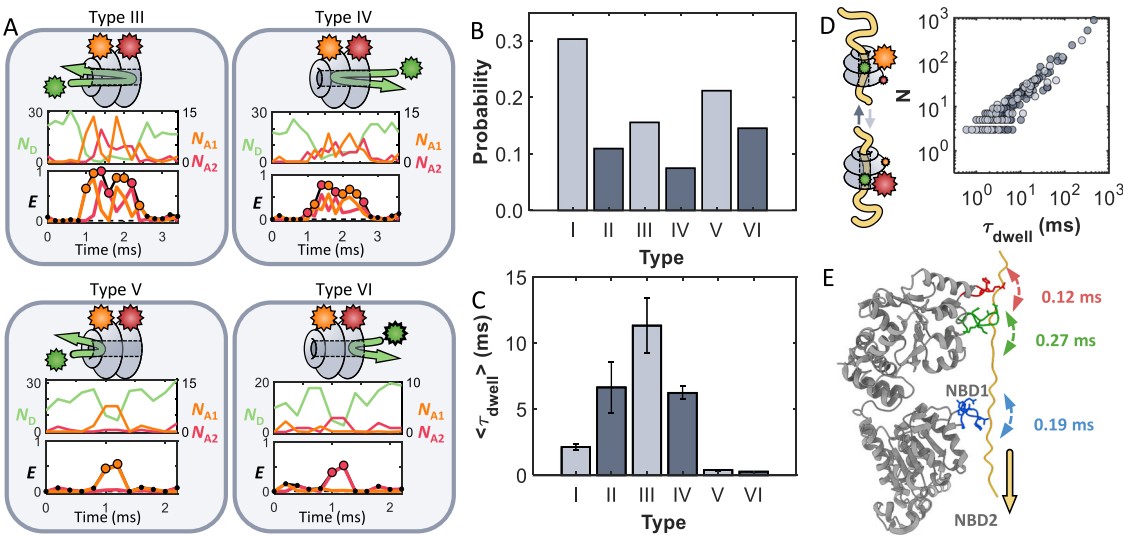

**Fig. 5 | ClpB exhibits partial translocation events. A** Sample incomplete translocation events, donor (green, $N_D$) and acceptors (acceptor 1: orange, $N_{A1}$, and acceptor 2: red, $N_{A2}$), and apparent FRET efficiency ($E_{A1}$, $E_{A2}$, and overall apparent efficiency as the black trace $E_{app}$) are provided. Type III (forward) and IV (backward) partial translocation events are characterized by casein reaching both NBDs but exiting from the same side it entered from. Type V (forward) and VI (backward) fleeting events are characterized by an interaction with only a single NBD. **B** Probability of occurrence of the different translocation events. From Type I to VI, the number of events are 438, 158, 225, 108, 306, and 210 for the 135 individually measured ClpB molecules. **C** Average time for each event type (Supplementary Fig. 20). Data are presented as mean values +/− SD, values are reported in Supplementary Table 6. **D** Correlation between the number ($N$) of back-and-forth motions and the dwell times for type III (light) and IV (dark) events. The correlation coefficient of $R = 0.98$ demonstrates that the long dwell times are directly proportional to $N$. The cartoon inlay illustrates the back-and-forth motions of casein. (**E**) Schematic of the previously determined sub-millisecond dynamics of the pore loops (PLs) in the presence of ATP and casein[56]. The three PLs (red, green, and blue) on each subunit directly engage the substrate and rapidly move up-and-down (PDB 6OAX[72]). These fast dynamics modulate the energy landscape between two (or more) rapidly interconverting states. ATP allosterically regulates the dynamics and conformational states of the PLs[56], generating a landscape in which the substrate preferentially diffuses forward from NBD1 towards NBD2.

Fig. 20, Supplementary Table 2). As expected, the addition of ATPγS significantly reduced the number of observed events and generated a 20-fold increase in dwell time (Supplementary Fig. 21), demonstrating that the three-color experiments were sampling the same rapid ATP-dependent translocation events. However, with the addition of the second dye, we now had the ability to further dissect the results, providing comprehensive detail to the picture of translocation.

Three-color FRET experiments uncovered a total of six types (I-VI) of translocation events. As shown in Fig. 4, translocation events often took place in the expected, forward direction (type I, starting at the N-terminus and proceeding to the C-terminus). Representative type I events in Fig. 4B, left, and Supplementary Fig. 22 demonstrate that the FRET events begin with acceptor 1 (orange) and terminate with acceptor 2 (red), indicating that casein has completely passed through ClpB. An exponential fit of the $E_{app}$ dwell time distribution of type I events revealed a $\langle \tau_{dwell} \rangle$ value of $2.1 \pm 0.2$ ms (Fig. 4d). While this value was very similar to the $\langle \tau_{dwell} \rangle$ values found in the two-color experiments, the three-color experiments confirm that complete translocation indeed occurs in such a short timeframe. Based on the placement of the two dyes on ClpB, the $\langle \tau_{dwell} \rangle$ values for translocation correspond to a lateral motion of 11.6 nm (which is the distance in Fig. 4A where $\varepsilon_{calc} > 0.5$), translating into an overall average translocation velocity of 14 $aa$/ms in the forward direction. Based on this estimate, the 190 $aa$ long casein, corresponding to a contour length of ca. 76 nm, would take only ~14 ms to be fully translocated through the lumen of ClpB. While not directly observed, both bulk stopped-flow mixing experiments[22] and single-molecule optical tweezer assays[18] support the existence of such fast translocation events. Although the short range of FRET limits our ability to rule out a combination of translocation steps and pauses, as proposed by single-molecule optical tweezer assays[18,42,43], the lack of intermediate FRET efficiency values in our data points towards a continuous translocation mechanism. In fact, sustained rapid translocation with speeds greater than 2.8 nm/ms

(equivalent to ~ 7 $aa$/ms) has been observed in DNA-translocating AAA+ proteins[44].

In addition to forward translocation events, there were also events that proceeded in the unanticipated backward direction (type II, starting at the C-terminus and continuing to the N-terminus, Fig. 4B, and Supplementary Fig. 23). Of all complete translocation events, 25% occurred in the backward direction, where casein entered ClpB from the NBD2 side (generating fluorescence in the second acceptor, marked red), moved past all three sets of pore loops, and exited through the NTD (generating fluorescence in the first acceptor, marked orange). While it has been assumed so far that translocation occurs only in the forward direction, this study not only reveals the existence of *retrograde* threading events but also demonstrates that they occur in relative abundance, highlighting their significance in the translocation activity of ClpB. These type II events differed from their type I counterparts in their $\langle \tau_{dwell} \rangle$ value of $6.6 \pm 1.9$ ms (Supplementary Fig. 20), which was about three times slower, pointing to a preferential forward bias in motion through ClpB's lumen.

## Partial threading events

In addition to complete forward and backward translocation, the three-color smFRET experiments also identified partial translocation events. Types III and IV events in Fig. 5A (also in Supplementary Fig. 24) correspond to incomplete translocation events where casein reached deep into the lumen, arriving at both NBDs and thus exhibiting energy transfer to both acceptors. However, the signals began and terminated with the same acceptor, demonstrating that casein entered and exited from the same side of ClpB. While partial threading events have been inferred from previous studies[19,45], we directly confirm their existence, also noting that they occur in a bidirectional manner. As in the case of type I and II events, a notable feature is that the forward type III events were significantly more abundant than backward type IV events (Fig. 5B). Type III events typically took $11.3 \pm 3.1$ ms, over 5 times longer

than the type I events (Fig. 5C). However, the backward type IV events, with a $\langle\tau_{\text{dwell}}\rangle$ of $6.2 \pm 0.5$ ms, lasted for nearly identical times as the backward type II complete translocation events. Another interesting feature is that the relatively long dwell times of both type III and IV events appear to originate from rapid back-and-forth motions of casein, as seen in the frequent interconversion between the two acceptors (orange and red) in the sample trajectories given in the top of Fig. 5A. Such a feature suggests that the long dwell times of these events are related to back-and-forth motions of the substrate in the lumen before exiting. A correlation plot between $\tau_{\text{dwell}}$ and the number of back-and-forth motions revealed a correlation coefficient of $R = 0.982$ (Fig. 5D).

Lastly, type V (forward) and VI (backward) events (Fig. 5A and Supplementary Fig. 25) were attributed to casein superficially binding, but not entering the lumen, resulting in the appearance of energy transfer to only one of the acceptors. Both type V and VI events exhibited very brief dwell times of under 0.5 ms (Fig. 5C), suggesting that these events represent instances where casein attempts to enter the lumen but fails to make it past one of the NBDs (which would result in energy transfer to both acceptors) before dissociation.

Overall, only two-thirds of all events started at the *N*-terminus of ClpB, while the rest started at the *C*-terminus. It is possible that in the presence of ClpB's co-chaperones, such as DnaK and DnaJ, which are typically required for aggregate recognition, a larger fraction of translocation events would originate from the *N*-terminal side[4]. However, even without these co-chaperones, the NTD binding groove on ClpB is known to bind substrates such as casein[46], and therefore likely contributes to the observed two-fold increase in probability for events that begin at the *N*-terminus.

In the presence of ATPγS, we found that there was a five-fold reduction in the number of events per ClpB molecule, similar to the four-fold reduction in interactions observed in the two-color experiments. Fitting the overall cumulative distributions of dwell times (Supplementary Fig. 26), it was found that ATPγS increased the overall dwell time by a factor of 20 compared to that of ATP. The lower factor than the one found in the two-color and fluorescence anisotropy experiments (~30) can be attributed to premature photobleaching of the donor before substrate disengagement due to the increased laser power used in these experiments. Despite the limited number of only 135 events (across 110 ClpB molecules) registered in the presence of ATPγS, we nonetheless found that ClpB exhibits all six types of events. However, the fractions of forward and backward events were nearly equivalent to each other in all cases, indicating that with ATPγS, the forward vs. backward bias vanishes. Importantly, this result demonstrates that ATP is necessary to rectify translocation in the forward direction. A complete summary of the frequency of each event type, along with their corresponding dwell times, can be found in Supplementary Tables 6 and 7. We also note that the strict nucleotide dependence and the orderly sequence of FRET signals observed in the three-color experiments further confirm that we are monitoring genuine substrate translocation events through the ClpB lumen, rather than via side entry or nonspecific interactions at the protein surface.

## Discussion

### ClpB drives diffusive-like translocation

The translocation activity of ClpB demonstrated in this work is surprisingly rich and very different than expected based only on structural studies of AAA+ machines. Translocation events are short and demonstrate a preferred directionality. These characteristics are abolished in the presence of ATPγS, indicating significant dependence on ATP. The bidirectional nature of translocation suggests that the energy landscape throughout the translocation process must be shallow. Indeed, our temperature study revealed an activation energy of only ~ 2 $k_B$T. The experiments further demonstrate that ClpB

facilitates both partial and complete translocation events in a bidirectional manner, albeit with a significant preference for forward motion.

We can gain deeper insight into the landscape by considering the dwell times and frequency of the different types of events. First, we find that the complete forward translocation events are, on average, over five times faster than the incomplete or backward translocation events. Based on this ratio, transition state theory[47] suggests that the difference in free energy between the forward and backward threading directions ($\Delta\Delta G = \Delta G_{\text{forward}} - \Delta G_{\text{backward}} = -\ln(\langle\tau_{\text{dwell}}\rangle_{\text{forward}} / \langle\tau_{\text{dwell}}\rangle_{\text{backward}})$) is no greater than 1 – 2 $k_B$T. Second, considering the scenario where casein has entered the lumen (i.e. not including the superficial types V and VI events), we find that casein is only slightly more likely to exit past NBD2 (Pr('*exit* via *NBD2*') = 0.59, types I and IV) than past the NTD (Pr('*exit* via *NTD*') = 0.41, types II and III). The Curtin–Hammett principle[48] suggests that the difference in transition-state energies ($\Delta\Delta G^{\ddagger} = -\ln(\text{Pr}('exit \text{ via } NBD2')/\text{Pr}('exit \text{ via } NTD'))$) between the forward and backward directions is less than 1 $k_B$T.

Our experiments paint a picture of a shallow landscape where, at any given time, casein is nearly as likely to diffuse in either direction. Further evidence for diffusive-like translocation arises from the type III and IV events, which exhibit hallmarks of rapid back-and-forth motions of casein within the lumen and show a direct correspondence between the number of these motions and the duration of the events. Notably, we observe that diffusion must occur quickly, resulting in translocation within just a few milliseconds. Remarkably, limiting the supply of ATP did not significantly alter the translocation time but rather reduced the frequency of translocation events, indicating that ATP must regulate substrate recruitment in addition to translocation directionality and velocity. Moreover, we find that there are, on average, ~ 1.5 events per hydrolysed ATP. Based on the 41% probability of complete threading, we calculate that each complete threading event correlates with the consumption of only $1.65 \pm 0.29$ ATP molecules. Together, these features suggest a mechanism where the utilization of just one or two ATP molecules is sufficient to drive rapid bidirectional translocation events. A fully detailed understanding of this feature requires knowledge about the average dwell time of ATP and its hydrolysis products on all 12 binding sites of ClpB, which is lacking at this point in time. It is important to note that ClpB activity may involve both releasing a protein from an aggregate and the subsequent translocation through the central pore, potentially requiring distinct mechanisms of action[1,22]. Our experiments with the intrinsically disordered casein focus on the latter and isolate the translocation process in the absence of any potential rate limitation due to the disaggregation step. While our study does not directly probe substrate unfolding, we emphasize that translocation is an essential and indispensable component of the overall disaggregation mechanism.

### ClpB as a Brownian motor

Two mechanisms have been proposed for molecular machine driven translocation: the *power stroke* and the *Brownian motor*[12,13,39,49–51]. While these two mechanisms can share some nuanced characteristics[49], they are classically quite different from one another. The *power-stroke* mechanism involves significant, fuel-driven conformational changes in the machine that generate irreversible translocation steps along a downhill free energy landscape. In contrast, a *Brownian motor* does not depend on large, coordinated motions. Instead, substrate movement is driven by thermal noise (Brownian motion), while an external energy source introduces some form of asymmetrical changes within the free energy landscape. This asymmetry restricts diffusion to one direction, resulting in directed motion. The mechanistic features utilized by ClpB discussed above align more closely with those expected of a *Brownian motor*[12,39]. Notably, the rapid translocation by ClpB is more akin to findings on transmembrane pores (e.g., α-hemolysin[52] and Cytotoxin-K[53], both with a similar 2 nm pore diameter to ClpB), though in these

cases, electric potential differences drive biomacromolecules across a membrane. We note that in our experiments, ClpB's central pore (-13 nm in length) is much shorter than the full contour length of casein (-76 nm), and casein is labeled on a central cysteine. Consequently, the early stages of threading, where work against the entropy of casein is expected to be greatest and therefore may slow translocation, have limited resolution. However, if entropic effects were a major hurdle, we would expect long dwell times at low FRET efficiencies (E ~ 0.2, based on Figs. 1C and 4A), due to binding at the entrance of ClpB's pore, which are absent in our data. Instead, we find that dwell times longer than 10 ms are rare (<15%) and, when present, exhibit FRET efficiencies similar to those of the rapid events. While our labeling strategy enables direct observation of substrate motion within the lumen, the dye placements may not fully capture weak or transient binding events at the NTD or during the very earliest threading stages, which could occur below our detection threshold.

A Brownian motor mechanism might be a general feature of AAA+ translocating proteins. In these proteins, a dominant entropic contribution to translocation should arise from the loss of conformational freedom of the disordered/unfolded translocated chain as it enters the narrow lumen, generating an initial entropic barrier to threading. However, as translocation progresses and the leading portion of the chain exits the pore, this loss of freedom is gradually alleviated, facilitating forward motion. Thus, while an entropic barrier (expected to be on the order of ca. 9 kBT[54]) may hinder binding or the earliest threading steps, we have no indication that it significantly impacts the overall translocation time, though it may lower the overall number of interaction events. An additional contribution involves the shape of its pore, including the substrate-engaging pore loops (PLs), which line ClpB's lumen; such geometric factors, combined with diffusion bias, were shown to generate entropic transport[55].

Our findings do not exclude the possibility of subunit motions proposed by structural models; however, they do demonstrate that these motions alone cannot fully account for the translocation mechanism. This raises a fundamental question: How does ClpB harness the energy from ATP to drive directional translocation? Our earlier studies[56,57] revealed that the substrate-engaging pore loops (PLs), which line ClpB's lumen, exhibit sub-millisecond up-and-down motions (Fig. 5E), suggesting a dynamic interaction that may regulate substrate translocation. Importantly, these stochastic ultrafast PL dynamics persist under all nucleotide conditions and even in the presence of casein. However, the populations of the PL conformational states are allosterically regulated by the nucleotide, leading to nucleotide-dependent asymmetries in their orientations[56]. Within this framework, we propose a ratcheting mechanism in which the substrate encounters an energy landscape with a preferred direction generated by PL dynamics. The rapid and asymmetric interconversion between PL states facilitates the rapid motion of casein within the lumen while rectifying translocation in the forward direction. This interpretation is consistent with prior findings on the AAA+ protease FtsH, where thermal motions were shown to drive subunit dynamics, and ATP acted to bias the conformational probability distribution[58]. Together, these observations reinforce the view that ATP functions not as a direct power source for large conformational strokes, but rather as an asymmetry generator that rectifies Brownian motion into productive translocation.

To explore whether the previously observed sub-millisecond dynamics of the PLs correlate with the millisecond translocation events observed in this study, we calculated the average time and amplitude (of motion) for each of the three sets of pore loops to complete their up-and-down motions in the lumen (Supplementary Table 8). As depicted in Fig. 5E, PLs 1, 2, and 3 take an average time of 0.12, 0.27, and 0.19 ms, respectively[56], to complete one up-and-down lateral motion of 1.4 – 1.7 nm[57]. By multiplying the time to complete one up-and-down 'cycle' by the amplitude of motion, we find that the

PL dynamics correspond to motions of 16 – 28 aa/ms (for PL2, (1.7 nm amplitude of motion) × (1 aa per 0.4 nm) / (0.27 ms per 'cycle')). These fast motions represent the expected limiting speed for a substrate traversing the rapidly evolving PLs and closely align with the ~ 10 aa/ms velocity estimated from the two-color experiments, as well as the 14 aa/ms velocity found for the type I events. In the presence of ATPγS, the PLs remain dynamic, but they lose their asymmetry[56], thereby abolishing the asymmetric landscape required for rapid directional translocation. Lastly, simulations demonstrate that rapid motions of asymmetrical restrictions in a pore, such as the up-and-down movements of the PLs, produce directional translocation as a result of entropic forces[59,60], supporting our model. It is noteworthy that entropically driven translocation is weakly dependent on temperature[55], which further supports our findings.

In conclusion, this study demonstrates that substrate translocation through the AAA+ machine ClpB occurs within milliseconds. ATP is essential for these rapid translocation events, with a limited supply of ATP reducing translocation frequency but not velocity, underscoring its role in substrate recruitment rather than direct force generation. Crucially, we reveal that translocation operates bidirectionally, with a 3:1 preference for forward threading. Substituting ATP with ATPγS abolishes directionality and hinders translocation, reinforcing ATP's essential role in regulating movement through ClpB's lumen. Our findings suggest that translocation cannot be exclusively governed by discrete power strokes following ATP hydrolysis. Instead, the observed diffusive-like behavior and low energy barriers associated with bidirectional motion support a mechanism akin to a *Brownian motor*, where ATP facilitates directionality rather than generating discrete steps. *Brownian motors*, which use an energy source to rectify diffusive motion[12], have been broadly studied as physical models for dissipative dynamics[61], yet direct observation of such motors in the context of biological nanomachines remains rare. Our results with ClpB may have broader implications for understanding other biological machines that couple ATP hydrolysis to substrate translocation, including protein disaggregases, unfoldases, and translocases across diverse cellular systems. By directly observing the interplay between translocation dynamics and ATP, this study establishes a framework for exploring mechanochemical coupling in other AAA+ proteins and beyond. Future studies using folded substrates will be important to directly probe the mechanochemical coupling between unfolding, translocation, and ATP hydrolysis. Such experiments would build on our current findings by revealing how force generation and substrate remodeling contribute to directional translocation in physiological contexts.

## Method

### Protein expression and purification

*Thermus thermophilus* ClpB with a six-residue histidine tag at its *C*-terminus was cloned into a pET28b vector. All ClpB mutants were generated by site-directed mutagenesis and standard cloning techniques and were verified by DNA sequencing. The wild-type (WT) and single mutant ClpB proteins (359 C and 771 C) were expressed and purified as previously reported[29]. *E. coli* BL21 (DE3) pLysS cells (Invitrogen) were transformed with the respective protein expression vectors and cultured at 37 °C until reaching an OD of 0.8. Protein expression was induced with 1 mM IPTG, after which the cultures were incubated overnight at 25 °C. Cells were harvested and the recombinant proteins purified using Ni−NTA resin (GE Healthcare), eluting with 250 mM imidazole. The eluates were then dialyzed overnight in the presence of ATP to remove imidazole. A double-mutant ClpB incorporated the unnatural amino acid *N*-ε-(prop-2-ynyloxycarbonyl)-*L*-lysine (ProK, Iris Biotech) at residue 771, and a cysteine at residue 176. ProK was coded with the Amber stop codon, allowing for co-transfection of the double mutant 176 C + 771TAG ClpB in pET28b together with pEvol PylRS/tRNApyl[62] (Addgene, Plasmid #127411, back

mutated to A346N/A348C) into *E. Coli* BL21 (DE3) pLysS cells (Invitrogen) using heat shock. Cells were recovered and incubated in 10 mL Luria Broth (LB, Thermo Fisher) media containing kanamycin (Sigma, 50 µg/mL) and chloramphenicol (Sigma, 25 µg/mL) for 16 h at 37 °C, 250 rpm. The overnight culture was used to inoculate 1 L of LB supplemented with corresponding antibiotics and arabinose (Formedium, 0.2 w/v%). After reaching an optical density (OD) of 0.5, the temperature was reduced to 25 °C, and ProK (2 mM) was added. After reaching an OD of 0.8, ClpB expression was induced by the addition of isopropyl ß-D-1-thiogalactopyranoside (Sigma, 1 mM) and incubated for 24 h. Purification was conducted in a similar manner as above. Briefly, cells were harvested and lysed using a French press, followed by heat shock treatment (80 °C, 10 min.). ClpB was recovered from the supernatant on a Ni Sepharose column (HisTrap FF, Cytiva) and dialyzed (50 kDa molecular weight cutoff, Gene Bio-Application Ltd.) overnight.

## Protein labeling and subunit mixing
Single-mutant ClpB was labeled with cyanine 3B maleimide (Cy3B, Cytiva) following a previously reported procedure[29]. κ-casein (Sigma) was labeled in a similar manner with either Cy3B or the cyanine-based dye 660 R maleimide (CF660R, Biotium). The probability of labeling casein (containing two native cysteines) with two dyes was minimized by reducing the dye ratio to 0.6 eq. per protein molecule (see *distribution of labeled casein inside liposomes* in the Supplementary Information for more details). Excess dye was removed using a desalting column (Sephadex G25 Superfine, Cytiva), followed by overnight dialysis (8 kDa, Gene Bio-Application Ltd.) against HEPES buffer (25 mM HEPES (J.T. Baker), 25 mM KCl (Sigma), 10 mM MgCl₂ (Sigma), pH 8). The ClpB double mutant was first labeled with Alexa Fluor 647 maleimide (AF647, Thermo Fisher), after which the unnatural amino acid was reacted with CF680R azide (Biotium) through copper-mediated click chemistry. Briefly, CF680R (4 eq.) was added to a solution of AF647 labeled ClpB (500 µL, 13 µM) in phosphate buffer (Sigma, 100 mM, pH 7). Next, CuSO₄ (Sigma, 200 µM) and tris-hydroxypropyltriazolylmethylamine (Lumiprobe, 1 mM) were added. The labeling reaction was then initiated with the addition of sodium ascorbate (Sigma, 5 mM) and the protein was left to react for 2 hours. Afterwards, the reaction was quenched by the addition of ethylene-diaminetetraacetic acid (EDTA, 2 mM). ClpB was separated from the reaction mixture using a desalting column (HiTrap, Cytiva). Any remaining copper or small molecules were removed by overnight dialysis (50 kDa) against a HEPES buffer containing EDTA (2 mM).

Hexameric assemblies of ClpB containing at most one single fluorescently labeled subunit were prepared by reassembling a mixture of labeled ClpB (1 eq.) with an excess (20 eq.) of wild-type ClpB. Homogeneous mixing of the subunits was ensured by performing the procedure in the presence of the guanidinium chloride (GdmCl, 6 M, Sigma) overnight. The denaturant was slowly removed by repeated dialysis (50 kDa) against HEPES buffers containing 4, 2, 1, 0.5, and 0 M GdmCl in four-hour intervals. The reassembled subunits were then dialyzed further against a HEPES buffer containing ATP (Sigma, 2 mM) overnight. The samples were then filtered (0.1µm, Whatman Anotop), aliquoted, flash-frozen, and stored at −80°C until further use. The low salt of the HEPES buffer (25 mM HEPES, 25 mM KCl, 10 mM MgCl₂, pH 8) and the persistence of nucleotide after reassembly ensured that ClpB remained in a hexameric assembly in all experiments[29,63,64]. Native gel demonstrating the reassembled ClpB is provided in Supplementary Fig. 5. The rotational freedom of the covalently bound dyes are quantified in the *fluorescence anisotropy decay experiments* section in the Supplementary Information.

## ATPase activity
The ATPase activity of ClpB was measured according to a previously published procedure[29] using a coupled colorimetric assay[65]. Briefly, ClpB (ca. 1 µM) was incubated in a buffer (25 mM HEPES, 25 mM KCl,

0.01% tween-20, pH 8) with an enzymatic ATP regeneration system (2.5 mM phospheonol pyruvate (Sigma), 10 units/mL pyruvate kinase and 15 units/mL lactate dehydrogenase (from rabbit, Sigma)), to which were added 1,4-dithioerythritol (2 mM, Sigma), EDTA (2 mM), nicotinamide-adenine dinucleotide (NADH, reduced form, 0.25 mM), and ATP (varied from 0 to 2 mM) at 25 °C. The reaction was started by the addition of MgCl₂ (12 mM). ATP hydrolysis was measured by monitoring the decrease of NADH absorption (340 nm) in time on a microplate reader (CLARIOstar, BMG Labtech). The ATPase activity of ClpB in the presence of casein (3 µM) and in its absence is shown in Supplementary Fig. 5.

## Surface-tethered lipid vesicle preparation
Following a previously published procedure[66], the fluorescently labeled protein molecules were encapsulated inside unilamellar vesicles, which were subsequently immobilized on a glass-supported lipid bilayer through a biotin-streptavidin linkage. Briefly, vesicles used for the formation of a glass-supported lipid bilayer were prepared from a lyophilized mixture (5 mg) of egg-phosphatidylcholine (Avanti) and 0.2 mol% 1,2-dioleoyl-sn-glycero-3-phosphoethanolamine-*N*-(biotinyl) (18:1 biotinyl PE, Avanti), which was reconstituted into a HEPES buffer (1 mL) by mixing. The suspension was extruded 75 times through a 0.1 µm filter (Whatman Anotop) to form unilamellar vesicles. The vesicle solution was inserted into a glass flow cell for 15 minutes, then rinsed with HEPES buffer, leaving behind a supported lipid bilayer on the interior surfaces of the cell. The cell was flushed with a solution of streptavidin (1 mg/mL), which attached to the biotinylated lipids. Vesicles containing ClpB and casein were prepared by reconstituting a lyophilized mixture (3 mg) of 1% mol biotinyl PE in 1,2-dimyristoyl-sn-glycero-3-phosphocholine (DMPC, Avanti) with a HEPES buffer solution containing ATP (2 mM), ClpB (200 nM), casein (3 µM), ascorbic acid (AA, 2 mM, Sigma) and protocatechuic acid (PCA, 1.5 mM, Sigma). The mixture was then extruded 75 times through a 0.1 µm filter to generate unilamellar vesicles with a ca. 120 nm diameter. Unencapsulated casein was removed by passing the liposome mixture through a MicroSpin S-400 HR column (Cytiva). Importantly, while the initial sample contained a 150-fold excess of casein, this ratio was carefully chosen so that the resulting liposomes would only contain a single molecule of casein and be sparsely populated with ClpB, leading to a near 1:1 ratio between the proteins in the ClpB-containing liposomes (see *Calculation of the expected number of proteins inside a liposome* in the Supplementary Information). The vesicle solution was diluted (1:100) and incubated inside the lipid-bilayer coated flow cell for 5 minutes to allow binding to the immobilized streptavidin. Unbound vesicles were removed by washing (3 × 200 µL) with the HEPES buffer containing ATP (2 mM, adjusted for experiments conducted at different concentrations). The HEPES buffer (100 µL, outgassed with $N_2$ to remove oxygen) with the addition of an oxygen-scavenging system[67,68] (1.5 mM PCA and ca. 5 nM protocatechuate 3,4-dioxygenase), a triplet quenching and radical scavenging system[68,69,] (2 mM AA, 1 mM methyl viologen) and ATP (2 mM, adjusted for experiments conducted at different concentrations) was added to the flow cell before sealing with silicone grease. Experiments in the presence of ATPγS (Roche, 2 mM) were prepared in a similar manner but were left to incubate with ATPγS for 40 minutes before reconstitution into liposomes.

## smFRET data acquisition
Single-molecule fluorescence data were acquired using a PicoQuant Microtime 200 confocal microscope. Two-color measurements were conducted in a continuous wave mode using a circularly polarized 561 nm laser with a power of ca. 400 nW (polarization and power were measured at the back focal plane of the objective (Olympus, UPlanFLN 100X/1.30 Oil Iris)). The positions of liposomes containing fluorescently labeled species were identified by raster scanning a piezo-driven stage (Physik Instrument, P-721.CLQ and P-733.2CL), and the

fluorescence of individual liposomes was monitored until photobleaching of the donor dye. The instrument was equipped with a ZT405/488/561/640rpcv2 (Chroma) primary dichroic mirror, the emission was split between the donor (Cy3B) and acceptor (CF660R) channels using a T635lpxr dichroic mirror (Chroma) and filtered using a 593/46 Brightline (Semrock) or a 731/137 Brightline bandpass filter (Semrock).

To increase the time resolution for the three-color experiments, we elected to perform the experiments under increased laser power (from 400 to 700 nW), which nevertheless still allowed obtaining fluorescence trajectories of a reasonable length. Three-color experiments were conducted using 40 MHz pulsed lasers with a sequence of one pulse of the acceptor laser (640 nm, ca. 50 nW) followed by three pulses of the donor laser (530 nm, ca. 700 nW). The excitation was circularly polarized, as measured at the objective. The positions of liposomes containing labeled ClpB were identified using an acceptor-only raster scan with a power of ca. 500 nW. The instrument was equipped with a TZ532/640rpc primary dichroic mirror (Chroma), and the emission was split into donor and acceptor by a T635lpxr dichroic mirror (Chroma). The acceptor emission was further split by a T685lpxr dichroic mirror (Chroma) into two acceptor channels (AF647 and CF680R). Before reaching the detector, the donor emission was filtered with a 582/75 Brightline bandpass filter (Semrock). The acceptor channels were filtered with a ET667/30 m or a ET720/60 m bandpass filter (Chroma) for the AF647 and CF680R channels, respectively.

## smFRET data analysis

The apparent FRET efficiency ($E_{app} = N_A / (N_D + N_A)$) of the two-color experiments was calculated using the photon counts (binned to 0.5 ms) from the acceptor ($N_A$) and donor ($N_D$) after corrections for background photons and channel crosstalk (i.e. acceptor emission detected in the donor channel and donor emission detected in the acceptor channel). Background flux was determined by the residual photon count in each channel after photobleaching of the donor, and channel crosstalk was found from trajectories of isolated donor- and acceptor-labeled molecules within liposomes. The acceptor had negligible crosstalk, while the donor had 13% of its emission appear in the acceptor channel. Further discussion on channel crosstalk can be found in the *derivation of leak and background corrections* and the *leak-factor determination* sections of the Supplementary Information. Manual filtering was used to identify trajectories containing both donor and acceptor fluorescence. Only trajectories showing clear anticorrelation between donor and acceptor intensities and a corresponding rise in FRET efficiency were included in the analysis. Translocation events in the trajectories were identified by peaks in the $E_{app}$ trajectories where $E_{app}$ exceeded 0.3. Since this method did not account for the rising and falling edges of the events, the edges were extended until the $E_{app}$ value was no longer statistically different (1.5 σ) than the local background values (using a sliding window of 7 bins). The dwell time $\tau_{dwell}$ of each event was taken as the length of each agglomerated region. $E_{peak}$ values were taken as the maximum $E_{app}$ value from the identified events.

While an explicit calculation of bona fide FRET efficiency values in multicolor experiments can be performed[70], for the purpose of this work, we elected to calculate apparent FRET efficiencies in simpler terms, directly using the experimentally obtained photon count rates of the donor ($N_D$) and the two acceptors ($N_{A1}$ and $N_{A2}$)[70,71]. The apparent FRET efficiencies in three-color experiments were calculated using the photon counts from 0.2 ms bins after background correction. Due to the relatively low photon counts (~ 10 – 30 photons/bin), applying additional correction factors such as channel crosstalk to the individual trajectories introduced a large amount of unwanted noise and, therefore, was avoided. Further discussion on the correction factors, including inter-acceptor energy transfer, can be found in the sections

on *leak-factor determination* and *determining inter-acceptor energy transfer* in the Supplementary Information. The apparent $E_{app}$ to each acceptor was calculated as $E_{Ai} = N_{Ai} / (N_D + N_{A1} + N_{A2})$, where Ai represents acceptor 1 or 2, and the total apparent FRET as $E_{app} = (N_{A1} + N_{A2}) / (N_D + N_{A1} + N_{A2})$. Following a similar procedure as in the two-color analysis, interaction events were identified using the total $E_{app}$ values, such that the extracted $\tau_{dwell}$ values represent the total observed dwell time regardless of which acceptor is more prominent. Once identified, each bin in an event was then assigned a color based on which acceptor was most prominent. While we elected not to apply spectral crosstalk corrections directly to the trajectories, acceptor 1 (AF647) had a moderately large fraction of its emission (37%) appear in the acceptor 2 channel (CF680R). As such, a bin was assigned to acceptor 2 if $E_{A2}$ contributed over 60% of the total $E_{app}$; otherwise, it was assigned to acceptor 1. The $E_{A2}/E_{app} > 0.6$ cut-off value was chosen because it corresponds to the situation where the donor is experiencing equivalent energy transfer to both acceptors (i.e., after correction for crosstalk, $\varepsilon_{A1} \sim \varepsilon_{A2}$).

## Event dwell-time characterization

The length of each event was characterized by the value $\tau_{dwell}$ as discussed in the *smFRET data analysis* section. To determine the exponent α of the power law distribution of $\tau_{dwell}$ values, $P(\tau_{dwell}) \sim \tau_{dwell}^{-\alpha}$, the survival probability that $\tau_{dwell}$ has a value greater than $\tau$, S($\tau$), was calculated for each point in the distribution, and Eq. 1 was fitted to the resulting data[30]. In this equation, $P$ is the probability distribution of $\tau_{dwell}$ values, $C$ is a normalization constant, and α is the power for $\tau_{dwell}$ values obeying the power law[30].

$$S(\tau) = \int_{\tau}^{\infty} P(t)dt = \frac{C}{\alpha - 1} \tau^{-(\alpha-1)} \qquad (1)$$

Distributions obeying power laws with α values less than two, as those reported herein, have a divergent mean and variance. Therefore, a representative overall average event time based on the collected statistics was found from the triexponential fit of the cumulative distribution (CDF, Eq. 2) of $\tau_{dwell}$ values.

$$CDF(\tau) = \int_{-\infty}^{\tau} P(t)dt \qquad (2)$$

In the presence of ATP, the short dwell times, representing the vast majority of events (see Supplementary Table 2), were characterized by an exponential fit of all events occurring in less than 10 ms. In doing so we noted that the few very long interactions did not spuriously skew the average dwell time reflecting the bulk of the short translocation events. Model parameters (and their corresponding standard errors) were optimized with the 'fitnlm' function in Matlab utilizing the Levenberg-Marquardt nonlinear least squares algorithm.

## Temperature-dependent smFRET experiments

In these experiments, the temperature inside the fluorescence cell was controlled by equipping the confocal microscope with a custom-built Peltier temperature unit, consisting of four Peltier elements controlling the temperature of an aluminum housing surrounding the cell, and another Peltier element controlling the temperature of an aluminum block directly coupled to the objective. The temperature inside the confocal volume was determined by measuring the change in the diffusion coefficient of Cy3B using fluorescence correlation spectroscopy (FCS). Further information can be found in *fluorescence correlation spectroscopy* and Supplementary Fig. 27 of the Supplementary Information. The constituents of the vesicular membranes were changed to lipids with transition temperatures close to those of the experiments (18:0 – 18:1 PC and 15:0 PC with transition temperatures of 6 and 35 °C, respectively), ensuring the membranes remained permeable to ATP

(Supplementary Figs. 28 and 29). Linear regressions and standard errors were computed using the method of ordinary least squares with the 'fitlm' function in Matlab.

## Fluorescence polarization experiments

The motional freedom of the encapsulated proteins was assessed using fluorescence polarization experiments[66]. Vesicles were loaded with either ClpB molecules fluorescently labeled with Cy3B or κ-casein labeled with CF660R and immobilized on a glass-supported lipid-bilayer surface (Surface-tethered lipid vesicle preparation). The dyes were excited with circularly polarized light to avoid photo selection, and the fluorescence was passed through a polarizing cube to split the beam into a horizontal ($I_H$) and vertical ($I_V$) component. The fluorescence polarization (P = $(I_V - I_H)/(I_V + I_H)$) was calculated for each 20 ms bin until photobleaching occurred, then averaged on a per molecule basis. The resulting polarization histograms in Supplementary Fig. 3 exhibit a very narrow distribution of polarization values centered about zero, demonstrating the ability of ClpB and casein to tumble rapidly within the liposome. For comparison purposes, ClpB and casein were non-specifically bound to a glass surface without passivation, resulting in significantly broader polarization distributions (Supplementary Fig. 3, with 3–5 times greater variance), which demonstrates the reduction in tumbling freedom of an immobilized protein.

## Steady-state fluorescence anisotropy

Steady-state fluorescence anisotropy measurements were performed on a Fluorolog-3 spectrofluorometer (Horiba) equipped with automated polarizers. Solutions of the fluorescently labeled proteins (ca. 200 nM) in HEPES buffer were measured in triplicate using a 5 s integration time. Samples containing Cy3B were excited at 550 nm and the emission was recorded at 570 nm. Samples containing CF680R were excited at 670 nm and the emission was recorded at 690 nm. Both the excitation and emission monochromators were set to slit widths of 5 nm. Anisotropy values were calculated using the built-in Fluor-Essence software. The fluorescence anisotropy values of Cy3B labeled casein in the presence of increasing ClpB concentration are reported in Supplementary Fig. 2. The fluorescence anisotropy of fluorescently labeled ClpB is provided in Supplementary Fig. 10. The alignment of the polarizers was verified by confirming that the anisotropy of light scattered by a dilute Ludox suspension (= $0.9887 \pm 0.0002$) was near unity.

## Reporting summary

Further information on research design is available in the Nature Portfolio Reporting Summary linked to this article.

## Data availability

All data generated and processed in this study have been deposited in the Zenodo database under accession code 17532905. A summary of the data in this manuscript is provided in the Supplementary Materials. Source data are provided with this paper. This study used the following PDB structures of ClpB: 6OAX and 1QVR. Source data are provided with this paper.

## Code availability

All code generated in this study has been deposited in the Zenodo database under accession code 17532905.

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

## Acknowledgements

We thank Drs. Pierre Goloubinoff, Hagen Hofmann, Axel Mogk, David Scheerer, Ilya Kuprov, and Michal Haran for reading and commenting on the manuscript. We thank Drs. Hagen Hofmann and Tanya Lasitza-Male for the use of their homebuilt Peltier control unit, Dr. David Scheerer for the fluorescently labeled enzyme adenylate kinase, and Dr. Hisham Mazal for his insightful input and kindly providing samples for initial experimental trials. This work was supported by a grant from the NSF-BSF program (no. 2021700, R.C., D.L., I.R., B.Y., G.H.). R.C. is grateful to the Azrieli Foundation for its generous funding of an Azrieli International Postdoctoral Fellowship.

## Author contributions

R.C., I.R., and G.H. conceptualized and designed the experiments. R.C. conducted the experiments. R.C., D.L., I.R., and Y.B. worked on protein expression, purification, and labeling. R.C. and G.H. analyzed the results and wrote the manuscript. All authors contributed to editing and discussions.

## Competing interests

The authors declare no competing interests.
