## [Peer Review file · Nature Communications]

A Stochastic Mechanism Drives Fast Substrate Translocation in the AAA+ Machine ClpB

Corresponding Author: Professor Gilad Haran

Version 0:

Reviewer comments:

Reviewer #1

(Remarks to the Author)

Review of the manuscript: "A Stochastic Mechanism Drives Fast Substrate Translocation in the AAA+ Machine ClpB"

The manuscript presents a compelling and technically sophisticated study that investigates the mechanism of substrate translocation by the AAA+ protein ClpB. Using single-molecule FRET (smFRET), the authors provide direct real-time evidence that ClpB translocates substrate proteins in a rapid, stochastic manner, challenging the prevailing power-stroke model and proposing a Brownian motor-like mechanism. The work is well-executed, conceptually novel, and of high relevance to the field of molecular machines and protein quality control.

Noteworthy Results

The authors demonstrate that ClpB translocates substrate proteins within milliseconds—approximately 1.6 ms per event—far exceeding the rate of ATP hydrolysis. This rapid translocation is shown to be ATP-dependent but not tightly coupled to hydrolysis events. The use of ATP γ S, a slowly hydrolysable analog, abolishes both rapid translocation and directionality, reinforcing the role of ATP in regulating substrate movement rather than directly powering it. Furthermore, the study reveals bidirectional translocation events with a 3:1 bias toward forward motion, and identifies partial threading events, suggesting a complex and dynamic interaction between ClpB and its substrates. These findings collectively support a Brownian motor-like mechanism, wherein ATP modulates the energy landscape to bias stochastic substrate diffusion through the ClpB pore.

Significance and Comparison to Established Literature

This work represents a significant advance in our understanding of AAA+ protein function. While previous structural studies and optical tweezer experiments have hinted at rapid translocation and partial threading, this manuscript provides direct molecular-level evidence for a stochastic mechanism. The findings challenge the traditional power-stroke model, which posits discrete, ATP-coupled conformational changes driving substrate movement. By contrast, the Brownian motor model proposed here aligns with theoretical frameworks but has rarely been observed in biological systems. The implications extend beyond ClpB to other AAA+ machines, including disaggregases, unfoldases, and translocases, making this study broadly relevant across molecular biology and biophysics.

Originality

The manuscript is original in both its experimental approach and its conceptual contributions. The use of two-color and three-color smFRET to dissect translocation dynamics in real time is innovative and provides unprecedented resolution. While the study builds on prior work (e.g., Mazal et al., 2021; Avellaneda et al., 2020), it offers new insights into the mechanistic basis of substrate translocation and the role of ATP in modulating directionality and recruitment.

Support for Conclusions

The conclusions are well-supported by the data. The authors employ a range of controls, including nucleotide analogs, temperature variation, and ATP concentration dependence, to validate their findings. The statistical treatment of dwell times, event frequencies, and FRET efficiencies is thorough and appropriate. The correlation between ATPase activity and translocation frequency further reinforces the proposed mechanism. While some aspects, such as ATP dwell times on ClpB binding sites, remain to be explored, the current evidence is sufficient to support the main claims.

Data Analysis and Interpretation

The data analysis is rigorous and clearly presented. Minor limitations, such as photobleaching in three-color experiments and simplified assumptions in FRET efficiency mapping, are acknowledged and do not undermine the overall conclusions. The interpretation is consistent with the results and thoughtfully contextualized within the broader literature.

Methodology and Reproducibility

The methodology is sound and meets the expected standards in the field. The experimental design is robust, and the techniques employed are state-of-the-art. The Materials and Methods section is comprehensive, and the supplementary

materials provide additional clarity. The level of detail is sufficient to allow reproduction of the experiments by other researchers.

Recommendation

I recommend this manuscript for publication, subject to minor revisions. The authors may consider expanding the discussion on the implications of the Brownian motor model for other AAA+ systems and clarifying the limitations associated with dye placement and early threading resolution. These additions would further strengthen an already excellent manuscript.

Reviewer #2

(Remarks to the Author)

Recommendation: Accept after revision (clarifications, additional information)

General evaluation

This study is a tour de force — real-time observation of fluorescence intensities of single molecules in a system where a supposedly floppy/disordered ~200 aa protein, κ -casein, binds to a “barrel”-like host structure (ClpB).

The authors claim that the measured fluorescence intensities monitor the passage of the floppy protein by measuring FRET between labeled casein and the outer regions of ClpB in dual- and triple-color FRET experiments. The presence of ATP and non- or slowly hydrolysable ATP analog ATP γ S modulate the kinetics of these fluorescence changes. It is a very valuable study, since classical structural biology techniques cannot distinguish between possible mechanisms here. Real-time monitoring is essential. Complementary biochemical and biophysical studies point towards a more stochastic and rapid mechanism.

I like very much how the paper is written. The equations follow the logical “cross-talk matrix”, clearer and more consistent than the “FRET community” papers. This is how FRET papers should be written.

1. Reproducibility and data openness

It is difficult to evaluate or test the claims based on the figures alone. I strongly recommend that the authors provide the primary photon data (not just processed trajectories) and the essential analysis scripts in a FAIR repository (e.g., Zenodo). Such transparency would (i) greatly facilitate verification of the claims, (ii) allow the community to reuse the results, and (iii) significantly increase the impact and credibility of the paper. Additionally, the “cherry” picking was / is not transparent, I would like that all recorded data / processed data are reported to be better able to judge the claims.

2. High claims versus experimental observables

The authors make far-reaching mechanistic claims. Essentially trying to “pin a floppy noodle in a hole” based on distance information alone. While single-molecule FRET is powerful, it cannot unambiguously resolve the path of a 200-aa disordered chain through a confined pore. Even less, if additional binding modes cannot be excluded. I am not sure that the data cannot be described by other binding modes (partial binding, processing, backfolding, etc.) additional simulations (coarse grained) of a string of beads could clarify what is going on, and suggest / rule out other possibilities of describing the intensity traces.

Specific concerns

- Relation between structure and observables: The connection between the structural model and the measured FRET efficiencies is not straightforward. Flexible dyes, linkers, and disordered polypeptide configurations can produce identical average FRET values. It is not clear how this degeneracy was treated.
- Local environment and quenching: The acceptor CF660R (a rhodamine) is a good choice, since unbridged Cy dyes would likely isomerize. However, rhodamines are prone to quenching by Trp (strong), Tyr, His, and Met (weaker). During translocation, the acceptor could transiently contact such residues, leading to intensity fluctuations unrelated to FRET. Were fluorescence lifetimes monitored? Any evidence from ALEX or lifetime heterogeneity analyses that quenching is negligible?
- Rotational averaging: Was the donor anisotropy low enough to justify assuming $\kappa^2 = 2/3$? The comparison between experimental E and calculated ϵ_{calc} implicitly assumes full rotational averaging, which might not apply in the lumen.
- Alternative explanations: At the high local concentrations inside vesicles (~2–3 μM), interactions between casein and the outer surface of ClpB could also alter anisotropy or yield apparent FRET. Similarly, if casein behaves as a floppy chain, parts of it could stick to the lumen wall or sample the outer ring without full passage, giving rise to similar transient FRET signals. Stiffening or loosening of the ClpB “barrel” could modulate these interactions.

Possible improvements

- Use of time-, polarization-, and color-resolved detection (four detectors with pulsed PIE excitation) would have allowed to simultaneous analysis of lifetime, anisotropy, and intensity and would make the conclusions less speculative.
- Alternatively, coarse-grained simulations of casein passage and the corresponding FRET distributions would provide a more molecular interpretation.

3. Simplified ϵ_{calc} model

The method used to compute ϵ_{calc} (Supplement p. 6 ff.) slices the ClpB structure into 150 planes and calculates distances from a fixed labeling site to the pore boundary. While elegant, it entirely neglects the flexibility of casein, the dyes, and the linkers. Because the ϵ_{calc} map is later used for biological conclusions, this simplification must be acknowledged, or ideally replaced by a coarse-grained model (e.g., AV or accessible-volume-based simulation). Otherwise, the comparison of E_{peak} with ϵ_{calc} is not physically meaningful — E_{peak} reflects time-averaged, conformationally averaged FRET; ϵ_{calc} is purely geometric and static.

4. Dwell-time and kinetic analysis

- Binning and detection limits: The dwell-time analysis depends on time binning; is it possible that even faster events were missed? Photon-count-based FCS on the same system could help to exclude faster dynamics (< 10 ms). In ATP-free measurements, the end of τ_{dwell} distributions is not reached, could the process be faster than inferred?
- Selection of traces: Many trajectories were recorded, but the selection procedure is unclear. Were traces chosen manually (“cherry-picking”) or by an algorithm? Which metrics defined valid events? Please provide primary data for all traces, indicate the rejection criteria, and explain how representative the shown examples are.

5. Thermodynamics and entropic aspects

The authors could comment on the entropic driving forces involved. Does casein translocation squeeze water from the lumen, thereby increasing overall system entropy? A rough back-of-the-envelope estimate (water cage volume vs. protein volume) could help rationalize the weak temperature dependence of the kinetics.

The observed weak temperature and ATP dependence is reminiscent of other systems, e.g. GTP-processing membrane-remodeling proteins, where ATP binding rigidifies the structure, facilitating motion, and hydrolysis adds irreversibility. A short comparison would clarify the conceptual link.

6. Specific textual and figure comments

- Figure legends are cut off in several places (main and supplement, e.g. Fig. S2 axis labels). Please correct before publication.
- Use proper averaging brackets $\langle \rangle$ instead of angle brackets $\langle \rangle$.
- The statement that τ_{dwell} represents a segment of ~18 aa assumes a freely disordered yet linearly extended chain; this assumption should be supported by references or simulations.
- The discussion of internal friction citing Soranno et al. (PNAS 2012) should be toned down: Soranno did not study casein, and interactions with ClpB could alter kinetics.
- Some parts of the supplemental description of FRET efficiency calculations contain typos (“Calculation of Calculation ...”).
- The cross-talk matrix equations are excellent and should remain; they exemplify best practice.

7. Minor observations and technical notes

- CF660R photophysics: verify that acceptor photophysics under confined conditions match expectations (lifetime, photostability).
- Possible improvement: perform control experiments with a static FRET pair immobilized in vesicles to benchmark photon statistics.
- Clarify whether the observed “fast events” in three-color FRET are single-pass events or repeated interactions of the same molecule.

Summary

The paper is of high scientific quality and addresses an exceptionally difficult problem with creative single-molecule methodology.

It is ready for publication once the authors:

1. Provide access to primary data and essential analysis scripts.
2. Clarify assumptions in the FRET-efficiency modeling and rotational averaging.
3. Discuss alternative photophysical and mechanistic interpretations.
4. Describe the event-selection procedure transparently.
5. Fix minor typographical and figure issues.

With these improvements, the study will become a definitive reference on stochastic substrate translocation in AAA+ machines. Moreover, I suggest to strongly separate the results and the discussion. Currently, the interpretation / results are very suggestive. The results “make sense” and are supported by the experiments, however there may be also other interpretations of the data.

Reviewer #3

(Remarks to the Author)

Single-molecule FRET spectroscopy was used to track casein movement through ClpB, a AAA+ chaperone that is involved in proteostasis and functions in protein disaggregation by unfolding proteins. The authors isolate ClpB and a model substrate, casein, within lipid vesicles and find that translocation events, require ATP and take milliseconds, much faster than ATP hydrolysis times. Using single molecule three-color FRET experiments, they find that translocation events can occur bidirectionally but are not always complete. Altogether the results indicate a fast, stochastic Brownian motor-like mechanism translocates casein bidirectionally through ClpB.

The work is significant because there is currently a lack of agreement in the chaperone field about the mechanism of unfolding and translocation of substrates through ClpB/Hsp104. Although it has been previously published that ClpB acts by a Brownian-ratchet mechanism by the Haran group, older reports suggested a power stroke or hand over hand movement. Thus, there is considerable interest in this question.

The manuscript is generally well-written, and the data presented support the conclusions drawn. However, there are many technical details presented in the main text that make it hard for a general reader to assimilate.

Points to be addressed:

1. The finding that casein likely moves through the ClpB pore by a Brownian motion is an interesting phenomenon. However, it is somewhat misleading to say that casein is a ClpB substrate, since it is an intrinsically disordered protein and not unfolded by ClpB. Rather, it is bound and released by ClpB. The study could benefit from the use of an additional substrate other than casein to demonstrate if translocation via Brownian motion is or is not unique to this substrate.
2. The important question in the field that is not addressed is the mechanism of translocation of substrate proteins that are unfolded by ClpB. Future studies will also have to contain a substrate that depends on ClpB and the Hsp70 system for unfolding and translocation.
3. It is known that casein translocation through ClpB involves substrate engagement by the pore loops, and this is facilitated by ATP hydrolysis. Are ClpB pore loop mutants defective in translocating casein and does ATP no longer facilitate directionality in these mutants? Are ATP binding and hydrolysis mutants defective in translocation using this assay?
4. Why was *T. thermophilus* ClpB used? Would the results be the same at the higher temperatures where ClpB T.th. is more

active? Would results be the same with a ClpB homolog, such as E. coli ClpB, that functions well at the temperatures used in the experiments.

5. Since it is known that ATP alone in the absence of Hsp70 and its cochaperones will not promote unfolding and translocation by ClpB, the translocation of casein by Brownian motion described here may not be physiologically important.

6. Would the combination of ATP and ATP γ S, a combination that is known to promote protein unfolding and translocation in the absence of Hsp70 and its cochaperones, give similar results?

7. What is the rate of ClpB substrate exchange? For the experiments to be valid, translocation has to be faster than ClpB substrate exchange.

8. Low concentrations of guanidine hydrochloride (GuHCl) stimulate bacterial ClpB ATPase activity and inhibit its disaggregating activity (Nowicki et al 2011 Cell Stress Chaperones). What are the effects of GuHCl on bidirectional substrate translocation via Brownian motion? As a separate question, the methods do not indicate how much GuHCl remains following dialysis to generate mixed hexamers.

Reviewer #4

(Remarks to the Author)

Version 1:

Reviewer comments:

Reviewer #1

(Remarks to the Author)

Thank you for the careful revision. The manuscript is now streamlined and compelling.

The expanded discussion convincingly situates a Brownian motor-like mechanism within the broader AAA+ context without overreach.

Limitations due to dye placement and earliest threading are clearly acknowledged and appropriately bounded.

Data, controls, and statistics remain rigorous; figures and SI are clear and sufficient for reproducibility.

The narrative is focused and the conclusions are well supported by the evidence.

I support publication.

Reviewer #2

(Remarks to the Author)

The authors have carefully addressed all points raised in my review. In particular:

They now provide full access to raw photon data and analysis scripts via Zenodo, which resolves my primary concern regarding reproducibility and data transparency.

Alternative interpretations and photophysical considerations (including quenching and rotational averaging) are adequately discussed.

The event-selection procedure is now transparent, and figure/text issues have been corrected.

The revised manuscript better separates results from interpretation, which improves readability.

Overall, the authors have implemented the requested clarifications without overstating their conclusions. The manuscript is technically and scientifically sound. The revisions address the outstanding points appropriately.

Recommendation: Ready for publication.

Reviewer #3

(Remarks to the Author)

The authors have adequately addressed the reviewers' comments.

Reviewer #4

(Remarks to the Author)

REVIEWER COMMENTS

Reviewer #1 (Remarks to the Author):

Review of the manuscript: “A Stochastic Mechanism Drives Fast Substrate Translocation in the AAA+ Machine ClpB”

The manuscript presents a compelling and technically sophisticated study that investigates the mechanism of substrate translocation by the AAA+ protein ClpB. Using single-molecule FRET (smFRET), the authors provide direct real-time evidence that ClpB translocates substrate proteins in a rapid, stochastic manner, challenging the prevailing power-stroke model and proposing a Brownian motor-like mechanism. The work is well-executed, conceptually novel, and of high relevance to the field of molecular machines and protein quality control.

Noteworthy Results

The authors demonstrate that ClpB translocates substrate proteins within milliseconds—approximately 1.6 ms per event—far exceeding the rate of ATP hydrolysis. This rapid translocation is shown to be ATP-dependent but not tightly coupled to hydrolysis events. The use of ATP γ S, a slowly hydrolysable analog, abolishes both rapid translocation and directionality, reinforcing the role of ATP in regulating substrate movement rather than directly powering it. Furthermore, the study reveals bidirectional translocation events with a 3:1 bias toward forward motion, and identifies partial threading events, suggesting a complex and dynamic interaction between ClpB and its substrates. These findings collectively support a Brownian motor-like mechanism, wherein ATP modulates the energy landscape to bias stochastic substrate diffusion through the ClpB pore.

Significance and Comparison to Established Literature

This work represents a significant advance in our understanding of AAA+ protein function. While previous structural studies and optical tweezer experiments have hinted at rapid translocation and partial threading, this manuscript provides direct molecular-level evidence for a stochastic mechanism. The findings challenge the traditional power-stroke model, which posits discrete, ATP-coupled conformational changes driving substrate movement. By contrast, the Brownian motor model proposed here aligns with theoretical frameworks but has rarely been observed in biological systems. The implications extend beyond ClpB to other AAA+ machines, including disaggregases, unfoldases, and translocases, making this study broadly relevant across molecular biology and biophysics.

Originality

The manuscript is original in both its experimental approach and its conceptual contributions. The use of two-color and three-color smFRET to dissect translocation dynamics in real time is innovative and provides unprecedented resolution. While the study builds on prior work (e.g.,

Mazal et al., 2021; Avellaneda et al., 2020), it offers new insights into the mechanistic basis of substrate translocation and the role of ATP in modulating directionality and recruitment.

Support for Conclusions

The conclusions are well-supported by the data. The authors employ a range of controls, including nucleotide analogs, temperature variation, and ATP concentration dependence, to validate their findings. The statistical treatment of dwell times, event frequencies, and FRET efficiencies is thorough and appropriate. The correlation between ATPase activity and translocation frequency further reinforces the proposed mechanism. While some aspects, such as ATP dwell times on ClpB binding sites, remain to be explored, the current evidence is sufficient to support the main claims.

Data Analysis and Interpretation

The data analysis is rigorous and clearly presented. Minor limitations, such as photobleaching in three-color experiments and simplified assumptions in FRET efficiency mapping, are acknowledged and do not undermine the overall conclusions. The interpretation is consistent with the results and thoughtfully contextualized within the broader literature.

Methodology and Reproducibility

The methodology is sound and meets the expected standards in the field. The experimental design is robust, and the techniques employed are state-of-the-art. The Materials and Methods section is comprehensive, and the supplementary materials provide additional clarity. The level of detail is sufficient to allow reproduction of the experiments by other researchers.

Recommendation

I recommend this manuscript for publication, subject to minor revisions. The authors may consider expanding the discussion on the implications of the Brownian motor model for other AAA+ systems and clarifying the limitations associated with dye placement and early threading resolution. These additions would further strengthen an already excellent manuscript.

We thank the reviewer for their highly positive and thoughtful assessment of our work. We are pleased that he/she found the study technically rigorous, conceptually novel, and of broad relevance to the AAA+ and molecular machine communities. We appreciate their suggestions for clarification, which we have addressed in the Discussion section under *ClpB as a Brownian motor* (pg. 14):

“...Instead, we find that dwell times longer than 10 ms are rare (<15%) and, when present, exhibit FRET efficiencies similar to those of the rapid events. While our labeling strategy enables direct observation of substrate motion within the lumen, the dye placements may not fully capture weak or transient binding events at the NTD or during the very earliest threading stages, which could occur below our detection threshold.

A Brownian motor mechanism might be a general feature of AAA+ translocating proteins. In these proteins, a dominant entropic contribution to translocation should arise from the loss of conformational freedom of the disordered/unfolded translocated chain as it enters the narrow lumen, generating an initial entropic barrier to threading. However, as translocation progresses and the leading portion of the chain exits the pore, this loss of freedom is gradually alleviated, facilitating forward motion. Thus, while an entropic barrier (expected to be on the order of ca. 9 kBT (56)) may hinder binding or the earliest threading steps, we have no indication that it significantly impacts the overall translocation time, though it may lower the overall number of interaction events. An additional contribution involves the shape of its pore, including the substrate-engaging pore loops (PLs, see below), which line ClpB's lumen; geometric factors combined with diffusion bias were shown to generate entropic transport (57)."

Reviewer #2 (Remarks to the Author):

Recommendation: Accept after revision (clarifications, additional information)

General evaluation

This study is a tour de force — real-time observation of fluorescence intensities of single molecules in a system where a supposedly floppy/disordered ~200 aa protein, κ -casein, binds to a “barrel”-like host structure (ClpB).

The authors claim that the measured fluorescence intensities monitor the passage of the floppy protein by measuring FRET between labeled casein and the outer regions of ClpB in dual- and triple-color FRET experiments. The presence of ATP and non- or slowly hydrolysable ATP analog ATP γ S modulate the kinetics of these fluorescence changes. It is a very valuable study, since classical structural biology techniques cannot distinguish between possible mechanisms here. Real-time monitoring is essential. Complementary biochemical and biophysical studies point towards a more stochastic and rapid mechanism.

I like very much how the paper is written. The equations follow the logical “cross-talk matrix”, clearer and more consistent than the “FRET community” papers. This is how FRET papers should be written.

We thank the reviewer for the detailed and insightful evaluation, as well as for recognizing the novelty, clarity, and rigor of our single-molecule FRET analysis. We have individually addressed the points below.

1. Reproducibility and data openness

It is difficult to evaluate or test the claims based on the figures alone. I strongly recommend that the authors provide the primary photon data (not just processed trajectories) and the essential analysis scripts in a FAIR repository (e.g., Zenodo). Such transparency would (i) greatly facilitate verification of the claims, (ii) allow the community to reuse the results, and (iii) significantly increase the impact and credibility of the paper. Additionally, the “cherry” picking was / is not transparent, I would like that all recorded data / processed data are reported to be better able to judge the claims.

We thank the reviewer for this valuable suggestion and fully agree that data transparency is critical for reproducibility and community impact. We would like to note that further examples of trajectories for each of the experiments were provided in the SI. Nevertheless, to facilitate independent verification and reuse, we have now deposited all raw and processed data, together with the analysis scripts, in a public Zenodo repository [[10.5281/zenodo.17532905](https://doi.org/10.5281/zenodo.17532905)]. The repository contains the following: Code: all MATLAB scripts used for data processing and analysis, Raw Data: single-photon and binned data of all trajectories in .mat format, and Processed Data: dwell-time and FRET-efficiency information for all trajectories classified as interaction events.

Installation and usage instructions for the software are included in the repository (summarized below). The software was developed in-house for this project and is not intended for commercial use. After data collection, trajectories were manually filtered following a stringent protocol to include only those exhibiting clear donor and acceptor fluorescence. The filtering protocol required clear anticorrelation between donor and acceptor intensity traces, accompanied by an increase in the FRET signal. Only trajectories meeting these criteria were retained. This step was necessary because a large fraction of liposomes contained donor-only signals. To ensure efficient sharing (due to the 50 GB file size limit), these non-acceptor trajectories were excluded from the uploaded dataset. However, the raw photon-level data for all these trajectories are fully available for independent inspection.

Briefly, the following analysis steps were performed with the deposited data. Photon data were converted from .ptu to .mat format and analyzed using the app “traj_GUI_v1”. Each trajectory was manually inspected to confirm the presence of both donor and acceptor channels. Background correction, photobleaching point determination, and FRET-peak assignment were carried out as described in the SI. For each trajectory, the dwell times and E_{peak} values were extracted and compiled into the “Processed Data” files. The MATLAB scripts used to generate all statistical plots and histograms in the main text are included in the repository for full transparency.

2. High claims versus experimental observables

The authors make far-reaching mechanistic claims. Essentially trying to “pin a floppy noodle in a hole” based on distance information alone. While single-molecule FRET is powerful, it cannot unambiguously resolve the path of a 200-aa disordered chain through a confined pore. Even less, if additional binding modes cannot be excluded. I am not sure that the data cannot be described by other binding modes (partial binding, processing, backfolding, etc.) additional simulations (coarse grained) of a string of beads could clarify what is going on, and suggest / rule out other possibilities of describing the intensity traces.

We agree with the reviewer that, given the intrinsic flexibility of a 200-aa unfolded polypeptide, our experiments cannot provide atomic-level resolution of the substrate position within the ClpB pore. Our goal was therefore not to reconstruct the detailed path of the entire chain, but to extract a robust, experimentally supported measure of translocation depth and temporal behavior of the substrate. It is important to note that our FRET data report on the position of a single labeled residue within the substrate, not the entire polypeptide chain.

Accordingly, we used the peak FRET efficiency as an approximate indicator of the relative progression of this residue along the translocation coordinate. This approach captures directional movement while remaining compatible with the temporal resolution of our measurements (0.5 ms), which is orders of magnitude slower than the intrinsic sub-microsecond dynamics of the disordered chain.

The strong ATP dependence of the observed rapid, directional events in the presence of ATP, and their complete abolishment with ATP γ S, demonstrates that these signals reflect active substrate motion rather than alternative static or binding modes. We therefore believe that our current data provide a direct and sufficient experimental basis for the mechanistic conclusions presented. In fact, binding events that may occur during substrate motion through the lumen of ClpB would not change the overall conclusions of the study.

While we agree that coarse-grained simulation (CGS) could, in principle, be a useful future addition to explore the conformational space and potential binding geometries of substrates within ClpB, we note that CGS depends entirely on the potentials used to describe interactions between system components. Therefore, CGS cannot be used to a priori discover events such as binding, which would appear depending on the strength of the implemented interaction between substrate elements and ClpB's elements. The same is true for other modes of translocation through ClpB's pore. We therefore propose that relying on the experimental results to provide information about translocation in ClpB is a reasonable approach. The experiment does not commit us to a specific mode of motion of substrates through the pore; rather, it provides information about an important observable, the rate of translocation, which can, in principle, be used to sift through different models. We leave such work to experts in computational studies.

Specific concerns

- Relation between structure and observables: The connection between the structural model and the measured FRET efficiencies is not straightforward. Flexible dyes, linkers, and disordered polypeptide configurations can produce identical average FRET values. It is not clear how this degeneracy was treated.

We agree with the reviewer that establishing a direct, quantitative connection between structural models and measured FRET efficiencies in a system involving flexible dyes, linkers, and a disordered polypeptide chain moving within ClpB is inherently complex. However, we do not interpret the FRET efficiency values in this work in a *quantitative* manner. Rather, we use the FRET efficiency values to report on the PROGRESS of the substrate chain through ClpB's lumen, for which these values only need to be used *only* in a *qualitative* manner. We have updated the main text (pg. 4) as follows.

“Given that the millisecond τ_{dwell} values are much longer than the timescale of the dynamics of an unfolded protein chain (10's–100's ns (33, 34)), there is ample time for the chain segment to explore the entire cross-sectional area of the pore during translocation. Even if interactions with ClpB slow these motions, they would remain orders of magnitude faster than our experimental time resolution of milliseconds. In addition, the flexibility of the dyes, linkers, and disordered chain further contributes to temporal and spatial averaging of the observed FRET efficiencies. We acknowledge that our ϵ_{calc} calculation, based on slicing the ClpB structure into planes and measuring distances from a fixed labeling site, does not explicitly account for the flexibility of casein, the dyes, or their linkers. Instead, ϵ_{calc} serves as a simplified geometric reference, providing an approximate measure of the average substrate position within the lumen rather than a fully quantitative structural model. For clarity, we present also ϵ_{calc} values corresponding to a chain passing through the center of the pore (red center line in Fig. 1C), which offers a consistent framework for comparing experimental and calculated FRET efficiencies without requiring detailed dynamic simulations.”

By adopting this simplification, we provide an approximate but consistent spatial reference for interpreting the FRET efficiencies, acknowledging that our approach does not capture all possible dye and linker conformations. Given the limited 0.5 ms temporal resolution of our measurements, this level of abstraction allows meaningful comparison of different events without overinterpreting the data. Furthermore, we note that the key information derived from the two-color experiments is the dwell time associated with substrate residence, rather than precise spatial mapping. Directionality and completeness of translocation events are then determined from the three-color experiments, which depend solely on which side of the pore the substrate enters or exits. Together, these complementary measurements provide robust mechanistic insight despite the inherent degeneracy from the disordered substrate.

- Local environment and quenching: The acceptor CF660R (a rhodamine) is a good choice, since unbridged Cy dyes would likely isomerize. However, rhodamines are prone to quenching by Trp (strong), Tyr, His, and Met (weaker). During translocation, the acceptor could transiently contact such residues, leading to intensity fluctuations unrelated to FRET. Were fluorescence lifetimes monitored? Any evidence from ALEX or lifetime heterogeneity analyses that quenching is negligible?

Although our main two-color experiments were performed with continuous-wave excitation (to maximize photon flux), we explicitly tested for acceptor quenching effects using separate pulsed interleaved excitation measurements, as shown in Figure S7 of the Supplementary Information. These data demonstrate that the acceptor remains equally active, regardless of whether it is undergoing a translocation event or not, with no detectable changes in acceptor brightness. Similar results are also observed for the three-color experiments (Figure S19), again showing constant acceptor signals throughout the trajectory. These results indicate that quenching effects are negligible under our experimental conditions and cannot account for the observed FRET fluctuations, which are instead generated by ATP-dependent substrate translocation. Another important point to note is that spikes appearing in the data always involve reversible anti-correlated intensity changes in the two channels, with the donor decreasing and the acceptor increasing, which can therefore be attributed to bona fide distance changes, rather than quenching.

- Rotational averaging: Was the donor anisotropy low enough to justify assuming $\kappa^2 = 2/3$? The comparison between experimental E and calculated ϵ_{calc} implicitly assumes full rotational averaging, which might not apply in the lumen.

We thank the reviewer for raising this important point regarding rotational averaging. We measured the steady-state anisotropy values for both donor and acceptor dyes, which are reported in Table S10. The obtained values are consistent with those reported in previous single-molecule FRET studies. Importantly, we note that time-resolved anisotropy (Table S11) reveals two tumbling times: a slow component corresponding to the overall rotation of the protein, and a fast component corresponding to the rapid tumbling of the linkers and dyes. Given the flexible linker connecting the acceptor dye to the substrate and the relatively large (~3 nm) diameter of the ClpB lumen, the dye is indeed expected to retain substantial rotational freedom. We also note that our main conclusions do not rely on explicit structural reconstruction or precise FRET efficiency values and FRET–distance conversion. Instead, the FRET efficiencies are used as approximate indicators of translocation depth, providing a representative and consistent framework for comparing different substrate states and nucleotide conditions.

- Alternative explanations: At the high local concentrations inside vesicles ($\sim 2\text{--}3\ \mu\text{M}$), interactions between casein and the outer surface of ClpB could also alter anisotropy or yield apparent FRET. Similarly, if casein behaves as a floppy chain, parts of it could stick to the lumen wall or sample the outer ring without full passage, giving rise to similar transient FRET signals. Stiffening or loosening of the ClpB “barrel” could modulate these interactions.

We thank the reviewer for raising these thoughtful points. Our observations suggest that transient interactions with the external surface of ClpB are not significant. Indeed, the rapid, ATP-dependent transitions we observe are abolished in the presence of ATP γ S, consistent with active translocation rather than passive binding. We therefore attribute the observed FRET changes primarily to substrate motion through the pore. Transient interactions between the substrate and the lumen wall are possible but would not alter the overall conclusions from the measurements. The occurrence of full or partial translocation events is deduced from the three-color experiments, which provide strong support for different modes of motion. Nonetheless, modulating the pore size to directly probe potential surface or lumen-wall interactions (if possible, without affecting the activity of ClpB) would indeed be a very interesting direction for future work.

Possible improvements

- Use of time-, polarization-, and color-resolved detection (four detectors with pulsed PIE excitation) would have allowed to simultaneous analysis of lifetime, anisotropy, and intensity and would make the conclusions less speculative.

We agree that combining lifetime, anisotropy, and intensity detection in a pulsed PIE/ALEX setup would provide valuable complementary information. Such measurements could be readily performed on our setup, and PIE was indeed used for the three-color experiments where separate excitation channels were a must. However, for the majority of our data, *continuous-wave excitation* offered clear advantages. Our goal here was to push the time resolution of the experiment using maximally possible photon fluxes and minimal photobleaching, which we found to be best achieved under *continuous-wave excitation*. By the same token, separating the signals into separate polarization channels would have compromised our time resolution, and we elected not to do that. We note that the control PIE experiments in the SI (Figures S7 and S19) demonstrate the stable activity of the dyes before photobleaching. Importantly, we also verified the modes of rotational diffusion of ClpB and its substrate using independent anisotropy experiments, as shown in Figure S3.

- Alternatively, coarse-grained simulations of casein passage and the corresponding FRET distributions would provide a more molecular interpretation.

We agree that coarse-grained simulations (CGS) could provide a useful molecular framework for interpreting the observed FRET distributions. Such simulations would indeed complement our

findings, and we view them as a promising next step to test and refine the stochastic Brownian translocation model presented here. However, as noted above, CGS depends on the choice of molecular potentials. Obtaining meaningful results would require a lengthy optimization of potentials, and it is not yet clear whether these efforts would yield substantially new mechanistic insight beyond what is already captured by our experimental observations.

3. Simplified ϵ_{calc} model

The method used to compute ϵ_{calc} (Supplement p. 6 ff.) slices the ClpB structure into 150 planes and calculates distances from a fixed labeling site to the pore boundary. While elegant, it entirely neglects the flexibility of casein, the dyes, and the linkers. Because the ϵ_{calc} map is later used for biological conclusions, this simplification must be acknowledged, or ideally replaced by a coarse-grained model (e.g., AV or accessible-volume-based simulation). Otherwise, the comparison of E_{peak} with ϵ_{calc} is not physically meaningful — E_{peak} reflects time-averaged, conformationally averaged FRET; ϵ_{calc} is purely geometric and static.

We agree that our ϵ_{calc} calculation, which slices the ClpB structure into planes and measures distances from a fixed labeling site, does not explicitly account for the flexibility of casein, dyes, or linkers. However, ϵ_{calc} is intended as a simplified geometric reference to provide an approximate measure of substrate position, rather than a fully quantitative structural model. We clarify this point with the addition of the following in *smFRET reveals ultrafast translocation of casein near the bottom of page 4*.

“... We acknowledge that our ϵ_{calc} calculation, based on slicing the ClpB structure into planes and measuring distances from a fixed labeling site, does not explicitly account for the flexibility of casein, the dyes, or their linkers. Instead, ϵ_{calc} serves as a simplified geometric reference, providing an approximate measure of the average substrate position within the lumen rather than a fully quantitative structural model. For clarity, we present ϵ_{calc} values corresponding to a chain passing through the center of the pore (red center line in Fig. 1C), which offers a consistent framework for comparing experimental and calculated FRET efficiencies without requiring detailed dynamic simulations.”

We emphasize that our main conclusions do not rely on the precise mapping of E_{peak} to absolute distances. Instead, E_{peak} is used as a time-averaged indicator of translocation depth, while kinetic and directional information comes from the dwell times and three-color FRET analysis. We agree that coarse-grained simulations might provide additional molecular insight and consider this a valuable avenue for future work.

4. Dwell-time and kinetic analysis

- Binning and detection limits: The dwell-time analysis depends on time binning; is it possible that even faster events were missed? Photon-count-based FCS on the same system could help to exclude faster dynamics (< 10 ms). In ATP-free measurements, the end of τ_{dwell} distributions is not reached, could the process be faster than inferred?

In our analysis, we used the smallest possible bin sizes of 0.5 ms for the two-color experiments and 0.2 ms for the three-color experiments. These bin sizes were chosen to maintain sufficient photon counts (at least ~ 10 – 30 photons per bin) for reliable detection of FRET transitions. While this allowed us to capture millisecond-scale translocation events robustly, we acknowledge that faster events below our time resolution could exist but cannot be resolved with the current photon statistics. The shape of the distributions, however, suggests that we pick up the major phenomenology with our measurements.

Regarding FCS, the translocation-related motion of the substrates in our experiments is diffusive. How to discriminate motion within ClpB from other diffusive motions, e.g., in free solution, is not a trivial task.

- Selection of traces: Many trajectories were recorded, but the selection procedure is unclear. Were traces chosen manually (“cherry-picking”) or by an algorithm? Which metrics defined valid events? Please provide primary data for all traces, indicate the rejection criteria, and explain how representative the shown examples are.

We have addressed this point in Comment 1 above, where we provide a description of the data selection protocol, inclusion criteria, and the availability of both the raw and processed datasets, along with the analysis code, on Zenodo [[10.5281/zenodo.17532905](https://doi.org/10.5281/zenodo.17532905)].

5. Thermodynamics and entropic aspects

The authors could comment on the entropic driving forces involved. Does casein translocation squeeze water from the lumen, thereby increasing overall system entropy? A rough back-of-the-envelope estimate (water cage volume vs. protein volume) could help rationalize the weak temperature dependence of the kinetics.

The observed weak temperature and ATP dependence is reminiscent of other systems, e.g. GTP-processing membrane-remodeling proteins, where ATP binding rigidifies the structure, facilitating motion, and hydrolysis adds irreversibility. A short comparison would clarify the conceptual link.

We agree that entropic effects are central to understanding polymer translocation through confined geometries. In the case of casein threading through ClpB, a dominant entropic barrier is expected to arise from the loss of conformational freedom of the disordered chain, and the shape of the pore, including the pore loops within it, is likely to contribute to the formation of a barrier as well (Reguera et al., Phys. Rev. Lett. (2006)). While water displacement within the lumen must

contribute, at least partially, to overcoming the entropic loss of casein itself, the relatively large pore diameter (ca. 3 nm) will limit the entropic gain of water release. As the leading segment of casein enters the narrow pore, the chain's configurational entropy is reduced, producing an initial entropic barrier that opposes translocation. However, as translocation progresses and the leading portion of the chain exits the pore, this loss of freedom is gradually alleviated, facilitating forward motion (Makarov, Acc. Chem. Res. (2009)). Importantly, the motion is biased by the investment of energy through ATP hydrolysis.

This framework explains the observed weak temperature dependence of the translocation kinetics. In contrast to activation over energetic barriers, where higher temperatures accelerate motion, entropically governed transport exhibits only minor temperature dependencies (Reguera et al., Phys. Rev. Lett. (2006)). The modest temperature dependence seen in our data is thus consistent with the scenario discussed above, in which ClpB rectifies largely thermally driven stochastic motions through ATP-dependent asymmetry rather than by overcoming a large barrier. In this view, ATP hydrolysis biases the fast, thermally fluctuating pore-loop dynamics to produce directional motion, while the chain's internal entropy facilitates completion of translocation once threading has begun.

To further clarify this, we modified the discussion in *ClpB as a Brownian motor* (pg. 14):
“In these proteins, a dominant entropic contribution to translocation should arise from the loss of conformational freedom of the disordered/unfolded translocated chain as it enters the narrow lumen, generating an initial entropic barrier to threading. However, as translocation progresses and the leading portion of the chain exits the pore, this loss of freedom is gradually alleviated, facilitating forward motion. Thus, while an entropic barrier (expected to be on the order of ca. 9 kBT (55)) may hinder binding or the earliest threading steps, we have no indication that it significantly impacts the overall translocation time, though it may lower the overall number of interaction events. An additional contribution involves the shape of its pore, including the substrate-engaging pore loops (PLs, see below), which line ClpB's lumen; geometric factors combined with diffusion bias were shown to generate entropic transport (57).”

6. Specific textual and figure comments

- Figure legends are cut off in several places (main and supplement, e.g. Fig. S2 axis labels). Please correct before publication.

We thank the reviewer for pointing out these issues, which we have now addressed.

- Use proper averaging brackets $\langle \rangle$ instead of angle brackets $< >$.

We are grateful to the reviewer for their careful reading. The proper brackets are now used.

- The statement that τ_{dwell} represents a segment of ~ 18 aa assumes a freely disordered yet linearly extended chain; this assumption should be supported by references or simulations.

The 18 aa length refers to the distance through ClpB's lumen in which $\epsilon_{\text{calc}} > 0.5$, which we calculated using the conversion of 0.4 nm per aa residue (Ainavarapu et al. Biophys. J. 2007). We have clarified this in the main text on pg. 4. *“Thus, τ_{dwell} values represent the time that it takes the labeled residue to traverse a segment of the ClpB pore corresponding to a length equivalent to ~ 18 amino acids (0.4 nm per amino acid) of the substrate (where $\epsilon_{\text{calc}} > 0.5$), and E_{peak} corresponds to the minimum proximity between the two labels during each translocation event, rather than the motion of the entire polypeptide chain.”*

- The discussion of internal friction citing Soranno et al. (PNAS 2012) should be toned down: Soranno did not study casein, and interactions with ClpB could alter kinetics.

We agree with the reviewer that the reference to Soranno et al. (PNAS 2012) should be interpreted with caution, as those measurements were performed on unfolded proteins in solution rather than casein interacting with ClpB. The cited timescale (10–100 ns) reflects the well-established dynamics of unfolded polypeptide chains in the absence of binding forces, as also discussed in other places, e.g., Krieger et al. J. Phys. Chem. B 2023. Nevertheless, even if interactions with ClpB slow these motions, they would remain orders of magnitude faster than our experimental time resolution of milliseconds.

- Some parts of the supplemental description of FRET efficiency calculations contain typos (“Calculation of Calculation ...”).

The SI has been proofread and corrected.

- The cross-talk matrix equations are excellent and should remain; they exemplify best practice.

We appreciate the acknowledgment and are pleased that the inclusion of the cross-talk matrix equations was found useful.

7. Minor observations and technical notes

- CF660R photophysics: verify that acceptor photophysics under confined conditions match expectations (lifetime, photostability).

We thank the reviewer for this point. We verified the photophysical stability of the acceptor under our experimental conditions through multiple controls, including fluorescence polarization measurements (Fig. S3), liposome occupancy experiments (Fig. S6 for the two-color setup and

Fig. S18 for the three-color experiments). These results confirm that the acceptor behaves as expected in confined conditions, with no indications of altered photophysics.

- Possible improvement: perform control experiments with a static FRET pair immobilized in vesicles to benchmark photon statistics.

We note that our lab has significant experience with vesicle encapsulation, e.g., Boukobza et al. J. Phys. Chem. B 2001 and Piwonski et al. PNAS 2012. In these studies, we have not seen an impact of encapsulation on photophysics. This is the expected behavior if the fluorescently labeled entity does not interact with vesicular walls. We verify this in the current case, Figure S3, by looking at the polarization of individual encapsulated molecules.

- Clarify whether the observed “fast events” in three-color FRET are single-pass events or repeated interactions of the same molecule.

We thank the reviewer for bringing this point to our attention. Each liposome was designed to contain a single fluorescently labeled casein molecule, so the observed events represent repeated interactions of the same substrate with ClpB, rather than multiple independent translocation events from different molecules. To clarify this in the manuscript, we have added the following to the beginning of the Results (pg. 3): “*Each liposome contained a single labeled casein molecule, and the fluorescence trajectory therefore reports multiple, repeated interaction events between the same casein and ClpB.*”. Importantly, however, these repeated interactions are separated by a significant amount of time and can therefore be considered independent of one another.

Summary

The paper is of high scientific quality and addresses an exceptionally difficult problem with creative single-molecule methodology.

It is ready for publication once the authors:

1. Provide access to primary data and essential analysis scripts.
2. Clarify assumptions in the FRET-efficiency modeling and rotational averaging.
3. Discuss alternative photophysical and mechanistic interpretations.
4. Describe the event-selection procedure transparently.
5. Fix minor typographical and figure issues.

With these improvements, the study will become a definitive reference on stochastic substrate translocation in AAA+ machines. Moreover, I suggest to stronger separate the results and the discussion. Currently, the interpretation / results are very suggestive. The results “make sense” and are supported by the experiments, however there may be also other interpretations of the data.

We thank the reviewer again for the kind remarks. We hope that our responses above satisfy all remaining open questions.

Reviewer #3 (Remarks to the Author):

Single-molecule FRET spectroscopy was used to track casein movement through ClpB, a AAA+ chaperone that is involved in proteostasis and functions in protein disaggregation by unfolding proteins. The authors isolate ClpB and a model substrate, casein, within lipid vesicles and find that translocation events, require ATP and take milliseconds, much faster than ATP hydrolysis times. Using single molecule three-color FRET experiments, they find that translocation events can occur bidirectionally but are not always complete. Altogether the results indicate a fast, stochastic Brownian motor-like mechanism translocates casein bidirectionally through ClpB.

The work is significant because there is currently a lack of agreement in the chaperone field about the mechanism of unfolding and translocation of substrates through ClpB/Hsp104. Although it has been previously published that ClpB acts by a Brownian-ratchet mechanism by the Haran group, older reports suggested a power stroke or hand over hand movement. Thus, there is considerable interest in this question.

The manuscript is generally well-written, and the data presented support the conclusions drawn. However, there are many technical details presented in the main text that make it hard for a general reader to assimilate.

We thank the reviewer for supporting the work and for pointing out multiple future experiments to be performed with ClpB. We would like to note that our work focuses on translocation through ClpB, a process that has been heavily debated. This is, of course, only one aspect of the protein's function, which disaggregates and unfolds proteins before translocating them through its lumen. Translocation is clearly very fast, and attempting to observe it simultaneously with the much slower unfolding reaction would be very difficult. The slow kinetics of substrate unfolding would likely obscure the much faster translocation step, making it extremely difficult to resolve the two processes simultaneously. This is precisely why we chose a loosely structured, soluble protein such as casein. Importantly, using a substrate that requires unfolding would have prevented us from observing ClpB's bi-directional translocation capability, a key feature highlighted in our study.

Beyond this issue, we would like to point out that calibrating the system for new studies with different substrates, particularly those that must unfold/disaggregate first, will be a lengthy process that will require modifying multiple elements of the experiment. In fact, the much longer time scales involved in these experiments will likely require a different single-molecule technique than used here. For example, one might need to use a camera-based setup that can track the fate of molecules over very long periods of time. We plan to conduct such studies and have recently submitted a grant proposal that outlines the methodology. However, we kindly propose that these studies remain for future work.

Below, we include additional answers to specific queries from the reviewer.

Points to be addressed:

1. The finding that casein likely moves through the ClpB pore by a Brownian motion is an interesting phenomenon. However, it is somewhat misleading to say that casein is a ClpB substrate, since it is an intrinsically disordered protein and not unfolded by ClpB. Rather, it is bound and released by ClpB. The study could benefit from the use of an additional substrate other than casein to demonstrate if translocation via Brownian motion is or is not unique to this substrate.

While casein is unfolded, it interacts well with ClpB, as has been shown by multiple studies. Indeed, as far as we can tell, *all structural studies of ClpB* involving a substrate made use of casein. Casein stimulates the ATP hydrolysis reaction of ClpB, and its interaction with the chaperone is therefore highly effective. (We further remind that the term Caseinolytic Protease, Clp, stems from the interaction of proteins of this family with casein, even those that are not human.)

2. The important question in the field that is not addressed is the mechanism of translocation of substrate proteins that are unfolded by ClpB. Future studies will also have to contain a substrate that depends on ClpB and the Hsp70 system for unfolding and translocation.

While we fully agree with the reviewer on the importance of studying ClpB function with folded substrates, we believe that due to the complexity of ClpB's activity, which involves both unfolding and translocation, it is essential to first dissect these steps and study them individually. Choosing casein, a substrate that bypasses the unfolding requirement, allowed us to isolate and characterize the translocation mechanism with clarity. This approach provides a necessary foundation for future studies involving more complex substrates. We plan to tackle the unfolding process itself in future studies, but, as noted above, this is a separate project with its own set of complexities.

3. It is known that casein translocation through ClpB involves substrate engagement by the pore loops, and this is facilitated by ATP hydrolysis. Are ClpB pore loop mutants defective in translocating casein and does ATP no longer facilitate directionality in these mutants? Are ATP binding and hydrolysis mutants defective in translocation using this assay?

Indeed, as multiple studies (including ours) have shown in the past, mutation of pore loops and ATP binding sites can modulate translocation. We know from our previous experiments that these are complex effects; therefore, we propose that they should also be explored in future studies.

4. Why was *T. thermophilus* ClpB used? Would the results be the same at the higher temperatures where ClpB *T.th.* is more active? Would results be the same with a ClpB homolog, such as *E. coli* ClpB, that functions well at the temperatures used in the experiments.

TT ClpB was selected because of its stability. We have shown that *TT* ClpB remains hexameric under the conditions of our experiments. However, we have also shown, as have many others, that

the protein is *highly active at ATP hydrolysis and protein substrate processing at room temperature*, making our studies highly relevant.

5. Since it is known that ATP alone in the absence of Hsp70 and its cochaperones will not promote unfolding and translocation by ClpB, the translocation of casein by Brownian motion described here may not be physiologically important.

While ATP alone does not promote unfolding/disaggregation, it does promote translocation, as we show here, and has been used in quite a few high-profile structural studies. Hsp70 is necessary for recruiting protein substrates to ClpB; however, since casein interacts with the chaperone even without Hsp70, it appears that it does not require it. Interestingly, as noted by the reviewer in the next point, substrate unfolding in the presence of the combination of ATP and ATP γ S also occurs in the absence of Hsp70. Considering the above, we wouldn't call the translocation observed here 'physiologically unimportant'.

6. Would the combination of ATP and ATP γ S, a combination that is known to promote protein unfolding and translocation in the absence of Hsp70 and its cochaperones, give similar results?

This is an interesting question that we should tackle in future studies. However, we note that it would be far from trivial to study mixtures of nucleotides on the single-molecule level. Such mixtures would lead to very heterogeneous results, as at each instance of time, a different number of ATP variants would be bound to ClpB. Interpreting such experiments would be quite demanding.

7. What is the rate of ClpB substrate exchange? For the experiments to be valid, translocation has to be faster than ClpB substrate exchange.

The rate of substrate exchange is the rate of interaction between ClpB and its substrates, as reported in Figure 3C of the manuscript. Indeed, it is significantly slower than translocation, but, surprisingly, it depends on the ATP hydrolysis rate.

8. Low concentrations of guanidine hydrochloride (GuHCl) stimulate bacterial ClpB ATPase activity and inhibit its disaggregating activity (Nowicki et al 2011 Cell Stress Chaperones). What are the effects of GuHCl on bidirectional substrate translocation via Brownian motion? As a separate question, the methods do not indicate how much GuHCl remains following dialysis to generate mixed hexamers.

The residual concentration of guanidine hydrochloride (GuHCl) after sample preparation is at most on the order of 10^{-9} M and cannot affect ClpB translocation. During our preparation, the protein was dialyzed against 0.5 M GuHCl and then dialyzed twice against a GuHCl-free buffer (500 mL)

with a sample volume of approximately 2 mL. The final dialysis step was performed overnight to ensure complete removal of the denaturant. During subsequent preparation, the reassembled ClpB was further diluted ~20-fold before encapsulation and an additional ~100-fold before immobilization. After immobilization, the sample was then again rinsed three times with an excess of buffer. Together, these steps reduce the final GuHCl concentration to vanishingly small levels, far below the millimolar concentrations known to influence ClpB activity (Nowicki et al., 2011).

Reviewer #4 (Remarks to the Author):

We thank the co-reviewer for their time and thoughtful evaluation of our manuscript.